# Can semi-supervised learning use all the data effectively? A lower bound perspective

**Alexandru Țifrea**\*
ETH Zurich
alexandru.tifrea@inf.ethz.ch

**Gizem Yüce**\*
EPFL
gizem.yuce@epfl.ch

**Amartya Sanyal**
Max Planck Institute for Intelligent Systems, Tübingen
amsa@di.ku.dk

**Fanny Yang**
ETH Zurich
fan.yan@inf.ethz.ch

## Abstract

Prior theoretical and empirical works have established that semi-supervised learning algorithms can leverage the unlabeled data to improve over the labeled sample complexity of supervised learning (SL) algorithms. However, existing theoretical work focuses on regimes where the unlabeled data is sufficient to learn a good decision boundary using unsupervised learning (UL) alone. This begs the question: Can SSL algorithms simultaneously improve upon both UL *and* SL? To this end, we derive a tight lower bound for 2-Gaussian mixture models that explicitly depends on the labeled and the unlabeled dataset size as well as the signal-to-noise ratio of the mixture distribution. Surprisingly, our result implies that no SSL algorithm improves upon the minimax-optimal statistical error rates of SL or UL algorithms for these distributions. Nevertheless, in our real-world experiments, SSL algorithms can often outperform UL and SL algorithms. In summary, our work suggests that while it is possible to prove the performance gains of SSL algorithms, this would require careful tracking of constants in the theoretical analysis.[1]

## 1 Introduction

Semi-Supervised Learning (SSL) has recently garnered significant attention, often surpassing traditional supervised learning (SL) methods in practical applications [5, 10, 22]. Within this framework, the learning algorithm leverages both labeled and unlabeled datasets sampled from the same distribution. Numerous empirical studies suggest that SSL can effectively harness the information from both datasets, outperforming both SL and unsupervised learning (UL) approaches [21, 42, 17, 25]. This observation prompts the question: how fundamental is the improvement of SSL over SL and UL? [2]

Prior theoretical results do not provide a consensus on this topic. One line of work demonstrates that under certain settings, SSL is capable of lowering the labeled sample complexity. In these settings, the unlabeled data possesses a significant amount of information about the conditional distribution (up to permutation). Some examples of this setting include distributions where unlabeled samples are enough to obtain small estimation error in mixture models [30, 18] or when the data is clusterable [31]. Therefore, despite SSL performing provably better than SL in these settings, it is only as good as UL (up to permutation). Another line of work challenges the above literature by identifying scenarios where SSL achieves the same error rate as SL. In these scenarios, even oracle knowledge about the

---

[1]Please consult the arxiv version of the paper for an updated manuscript.

[2]By error of UL we mean the prediction error up to sign. We formalize this paradigm of using UL first and then identifying the correct sign as UL+ in Section 2.2.

37th Conference on Neural Information Processing Systems (NeurIPS 2023).

marginal distribution fails to improve upon the error rates of SL algorithms since the marginal does not carry any information about the labeling i.e. the conditional distribution. Therefore, these settings do not allow SSL to improve upon SL but only upon UL.

In summary, the previous bounds do not provide a conclusive answer on the benefits of SSL; the positive and negative results consider different regimes dependent on sample sizes and the "compatibilibity" of the marginal distribution with the conditional distribution. In particular, they either improve upon UL or upon SL, but never both simultaneously. In this paper, we provide a first answer to the following question

*Can semi-supervised classification algorithms simultaneously improve*
*over the minimax rates of both SL and UL?*

Specifically, in Sections 2 and 3, we study this question in the context of linear classification for symmetric 2-Gaussian mixture models (GMMs) as done in several works in this domain [30, 18, 2, 24, 39]. In this setting, we derive minimax error rates for semi-supervised learning that specifically depend on the "regime", characterized by three quantities: the amount of available unlabeled data $n_u$, amount of available labeled data $n_l$, and the inherent signal-to-noise ratio (SNR) that quantifies the amount of information the marginal input distribution has about the conditional label distribution (see 3 for more details). An SNR-dependent minimax rate allows us to analyze the whole spectrum of problem difficulties for 2-GMMs. By contrasting the SSL minimax rates with established minimax rates for SL and UL, we find that in no regime can SSL surpass the statistical rates of both SL and UL. In conclusion, the optimal algorithm is the one that can adeptly switch between SL and UL algorithms depending on the regime and hence, it never uses the available data fully.

Nevertheless, statistical rates may not offer a complete picture for explaining the practical benefits of SSL algorithms. Several prevalent SSL algorithms, such as self-training, are more sophisticated than UL approaches and use labeled data not only for determining the sign but also for learning the decision boundary. In Section 4, we show that an SSL ensembling method and self-training [41, 7] can indeed improve upon the best of SL and UL algorithms even in proof-of-concept experiments for linear classification tasks on both synthetic and real-world datasets. Since the improvements cannot be captured by statistical rates, our results highlight the significance of the constant factors in future theoretical work that analyzes the advantage of SSL algorithms.

## 2 Problem setting and background

Before providing our main results, in this section, we first define the problem setting for our theoretical analysis, the evaluation metrics and the types of learning algorithms that we compare. We then describe how we compare the rates between SSL and supervised and unsupervised learning

### 2.1 Linear classification for 2-GMM data

**Data distribution.** We consider linear binary classification problems where the data is drawn from a Gaussian Mixture Model consisting of two identical spherical Gaussians with identity covariance and uniform mixing weights. The means of the two components $\boldsymbol{\theta}^*, -\boldsymbol{\theta}^*$ are symmetric with respect to the origin but can have arbitrary non-zero norm. We denote this family of joint distributions as $\mathcal{P}_{\text{2-GMM}} := \{P_{XY}^{\boldsymbol{\theta}^*} : \boldsymbol{\theta}^* \in \mathbb{R}^d\}$ that can be factorized so that the density reads $p_{XY}^{\boldsymbol{\theta}^*}(x, y) = p_{X|Y}^{\boldsymbol{\theta}^*}(x|y)p_Y(y)$ with

$$P_Y = \text{Unif}\{-1, 1\} \text{ and } P_{X|Y}^{\boldsymbol{\theta}^*} = \mathcal{N}(Y\boldsymbol{\theta}^*, I_d). \tag{1}$$

This family of distributions has often been considered in the context of analysing both SSL [30, 18] and SL/UL [2, 24, 39] algorithms. For $s \in (0, \infty)$, we denote by $\mathcal{P}_{\text{2-GMM}}^{(s)} \subset \mathcal{P}_{\text{2-GMM}}$ and $\Theta^{(s)} \subset \mathbb{R}^d$ the set of distributions $P_{XY}^{\boldsymbol{\theta}^*}$ and the set of parameters with $\|\boldsymbol{\theta}^*\| = s$, respectively. With this definition, we will be able to obtain refined bounds that depend explicitly on $s$. We consider algorithms $\mathcal{A}$ that take as input a labeled dataset $\mathcal{D}_l \sim \left(P_{XY}^{\boldsymbol{\theta}^*}\right)^{n_l}$ of size $n_l$, an unlabeled dataset $\mathcal{D}_u \sim \left(P_X^{\boldsymbol{\theta}^*}\right)^{n_u}$ of size $n_u$, or both, and output an estimator $\hat{\boldsymbol{\theta}} = \mathcal{A}\left(\mathcal{D}_l, \mathcal{D}_u\right) \in \mathbb{R}^d$. The estimator is used to predict the label of a test point $x$ as $\hat{y} = \text{sign}\left(\langle \hat{\boldsymbol{\theta}}, x \rangle\right)$.

| Learning paradigm | Excess risk rate | Estimation error rate |
|:---:|:---:|:---:|
| SL | $e^{-s^2/2}\dfrac{d}{sn_l}$ | $\sqrt{\dfrac{d}{n_l}}$ |
| UL(+) | $e^{-s^2/2}\dfrac{d}{s^3 n_u}$ | $\sqrt{\dfrac{d}{s^2 n_u}}$ |

Table 1: Known minimax rates of SL and UL for learning 2-GMMs [39, 24]. The minimax rates for UL are up to choosing the correct sign. This rate is the same as for UL+ if $n_l \geq \log(n_u)$. The notation $f(x) \asymp g(x)$ is equivalent to $f = \Theta(g)$.

**Evaluation metrics**    In this work, we consider two natural error metrics for this class of problems: prediction error and parameter estimation error[3]. For any vector $\theta$, we define its

> **Prediction error:** $\mathcal{R}_{\text{pred}}(\boldsymbol{\theta}, \boldsymbol{\theta}^*) := P_{XY}^{\boldsymbol{\theta}^*}(\text{sign}(\langle \boldsymbol{\theta}, X \rangle) \neq Y)$,    and (2)
>
> **Estimation error:** $\mathcal{R}_{\text{estim}}(\boldsymbol{\theta}, \boldsymbol{\theta}^*) := \|\boldsymbol{\theta} - \boldsymbol{\theta}^*\|_2$. (3)

When the true $\boldsymbol{\theta}^*$ is clear from the context, we drop the second argument for simplicity. In our discussions, we focus on the prediction error, but include a minimax rate for the estimation error for completeness. In particular, we bound the excess risk of $\boldsymbol{\theta}$, defined as the distance between the risk of $\boldsymbol{\theta}$ and the Bayes optimal risk

> **Excess prediction error:** $\mathcal{E}(\boldsymbol{\theta}, \boldsymbol{\theta}^*) := \mathcal{R}_{\text{pred}}(\boldsymbol{\theta}, \boldsymbol{\theta}^*) - \inf_{\boldsymbol{\theta}} \mathcal{R}_{\text{pred}}(\boldsymbol{\theta}, \boldsymbol{\theta}^*)$.

where $\inf_{\boldsymbol{\theta}} \mathcal{R}_{\text{pred}}(\boldsymbol{\theta}, \boldsymbol{\theta}^*)$ is achieved at $\boldsymbol{\theta}^*$ but can be non-zero.

For the set of all classification algorithms, we study the minimax expected error over a set of parameters $\Theta$. This worst-case error over $\Theta$ indicates the limits of what is achievable with the algorithm class. For instance, the minimax optimal expected excess error of the algorithm class over $\Theta$ takes the form:

> **Minimax excess error:** $\epsilon(n_l, n_u, \Theta) := \inf_{\mathcal{A}} \sup_{\boldsymbol{\theta}^* \in \Theta} \mathbb{E}[\mathcal{E}(\mathcal{A}(\mathcal{D}_l, \mathcal{D}_u), \boldsymbol{\theta}^*)]$. (4)

## 2.2    Minimax optimal rates of supervised and unsupervised learning

We distinguish between three kinds of learning that can be used in the SSL setting to learn a decision boundary $\hat{\boldsymbol{\theta}}$ but are designed to leverage the available data differently. For simplification, our discussion is tailored towards learning $\mathcal{P}_{\text{2-GMM}}$, though the ideas hold more generally.

**1) Semi-supervised learning (SSL)**    SSL algorithms, denoted as $\mathcal{A}_{\text{SSL}}$, can utilize both labeled $\mathcal{D}_l$ and unlabeled samples $\mathcal{D}_u$ to learn the decision boundary and to produce an estimator $\hat{\boldsymbol{\theta}}_{\text{SSL}} = \mathcal{A}_{\text{SSL}}(\mathcal{D}_l, \mathcal{D}_u)$. The promise of SSL is that by combining labeled and unlabeled data, SSL can reduce both the labeled and unlabeled sample complexities compared to algorithms solely dependent on either dataset.

**2) Supervised learning (SL)**    SL algorithms, represented by $\mathcal{A}_{\text{SL}}$, can only use the labeled dataset $\mathcal{D}_l$ to yield an estimator $\hat{\boldsymbol{\theta}}_{\text{SL}} = \mathcal{A}_{\text{SL}}(\mathcal{D}_l, \emptyset)$. The minimax rates of SL for distributions from $\mathcal{P}_{\text{2-GMM}}^{(s)}$ (see Table 1) are achieved by the mean estimator $\hat{\boldsymbol{\theta}}_{\text{SL}} = \frac{1}{n_l}\sum_{i=1}^{n_l} Y_i X_i$, for both excess risk and estimation error.

**3) Unsupervised learning (UL)**    Traditionally, UL algorithms are tailored to learning the generative model for marginal distributions. For $\mathcal{P}_{\text{2-GMM}}$ the marginal is governed by $\boldsymbol{\theta}^*$ and UL algorithms output a set of estimators $\{\hat{\boldsymbol{\theta}}_{\text{UL}}, -\hat{\boldsymbol{\theta}}_{\text{UL}}\} = \mathcal{A}_{\text{UL}}(\emptyset, \mathcal{D}_u)$ one of which is guaranteed to be close to the true $\boldsymbol{\theta}^*$. To evaluate prediction performance, we define the minimax rate of UL algorithms as the minimax rate for the closer (to the true $\boldsymbol{\theta}^*$) of the two estimators. This minimax rate of UL algorithms over $\mathcal{P}_{\text{2-GMM}}^{(s)}$ is known for both the excess risk and the estimation error [24, 39] (see Table 1). These rates are achieved by the unsupervised estimator $\hat{\boldsymbol{\theta}}_{\text{UL}} = \sqrt{(\hat{\lambda} - 1)_+}\,\hat{v}$, where $(\hat{\lambda}, \hat{v})$ is the leading eigenpair of the sample covariance matrix $\hat{\Sigma} = \frac{1}{n_u}\sum_{j=0}^{n_u} X_j X_j^T$ and we use the notation $(x)_+ := \max(0, x)$.

---

[3] For linear classification and a 2-GMM distribution from the family $\mathcal{P}_{\text{2-GMM}}$, a low estimation error implies not only good predictive performance, but also good calibration under a logistic model [29].

## 2.3 A "wasteful" type of SSL algorithm

Several SSL algorithms used in practice (e.g. Sim-CLR [9], SwAV [8]) follow a two-stage procedure: i) determine decision boundaries using only unlabeled data; and ii) label decision regions using only labeled data. We refer to this class of two-stage algorithms as **UL+** and denote them by $\mathcal{A}_{\text{UL+}}$. Early analyses of semi-supervised learning focus, in fact, on algorithms that fit the description of UL+ [30, 31].

---

**Algorithm 1:** UL+ algorithms $\mathcal{A}_{\text{UL+}}$

**Input :** $\mathcal{D}_l, \mathcal{D}_u$
$\{\hat{\boldsymbol{\theta}}_{\text{UL}}, -\hat{\boldsymbol{\theta}}_{\text{UL}}\} \leftarrow \mathcal{A}_{\text{UL}}(\mathcal{D}_u)$
$\hat{\boldsymbol{\theta}}_{\text{UL+}} \leftarrow$ Select from $\{\hat{\boldsymbol{\theta}}_{\text{UL}}, -\hat{\boldsymbol{\theta}}_{\text{UL}}\}$ using $\mathcal{D}_l$
**return** $\hat{\boldsymbol{\theta}}_{\text{UL+}}$

---

For linear binary classification, the two-stage framework is depicted in Algorithm 1. The following proposition upper bounds the excess risk and estimation error incurred by the UL+ estimator given by

$$\hat{\boldsymbol{\theta}}_{\text{UL+}} = \text{sign}\left(\hat{\boldsymbol{\theta}}_{\text{SL}}^{\top}\hat{\boldsymbol{\theta}}_{\text{UL}}\right)\hat{\boldsymbol{\theta}}_{\text{UL}} \quad \text{with } \hat{\boldsymbol{\theta}}_{\text{SL}} = \mathcal{A}_{\text{SL}}\left(\mathcal{D}_l\right). \tag{5}$$

**Proposition 1** (Upper bounds for $\hat{\boldsymbol{\theta}}_{\text{UL+}}$). *Let $\hat{\boldsymbol{\theta}}_{UL+}$ be the estimator defined in Equation* (5). *For any* $s \in (0, 1]$ *the following holds when* $n_u \geq (160/s)^2 d$ *and* $d \geq 2$:

$$\mathbb{E}\left[\mathcal{R}_{pred}\left(\hat{\boldsymbol{\theta}}_{UL+}, \boldsymbol{\theta}^*\right)\right] \lesssim \sqrt{\frac{d}{s^2 n_u}} + se^{-\frac{1}{2}n_l s^2 \left(1 - c_0\sqrt{\frac{d\log(n_u)}{s^2 n_u}}\right)^2}, \quad and$$

$$\mathbb{E}\left[\mathcal{E}\left(\hat{\boldsymbol{\theta}}_{UL+}, \boldsymbol{\theta}^*\right)\right] \lesssim e^{-\frac{1}{2}s^2}\frac{d}{s^3 n_u} + e^{-\frac{1}{2}s^2 n_l\left(1 - c_0\sqrt{\frac{d\log(n_u)}{s^2 n_u}}\right)^2}.$$

Appendix A contains the complete statement of the proposition including logarithmic factors and the proof. Note that for $n_l = o\left(\frac{1}{s^2}\log(n_u)\right)$, the first term in the upper bound dominates, thereby resulting in the same rate as the minimax rate of UL up to choosing the correct sign. We remark that, while Algorithm 1 defines UL+ algorithms as only using the unlabeled dataset $\mathcal{D}_u$ for the unsupervised learning step, one can also use the labeled dataset $\mathcal{D}_l$ without labels in that step. However, typically, in practice, UL+ style algorithms (e.g. SimCLR, SwAV) do not use the labeled data in this way, as they operate in a regime where unlabeled data is far more numerous than labeled data. Thus, we analyze Algorithm 1 in this work.

**Why UL+ algorithms are "wasteful"** As indicated in Algorithm 1, UL+ type algorithms follow a precise structure where labeled data is used solely to select from the set of estimators output by a UL algorithm. Intuitively, such algorithms do not take full advantage of the labeled data as it is not used to refine the decision boundary. In prior empirical and theoretical studies, this inefficiency has not been a problem since they have focused on the regime $n_u = \omega(n_l)$, where unlabeled data is often orders of magnitude more abundant than labeled data. When $n_u = \Theta(n_l)$ or even $n_u \ll n_l$, however, Proposition 1 shows how this two-stage approach of UL+ can become strikingly ineffective. For simplicity, consider the extreme scenario where $n_u$ is finite, but $n_l \to \infty$. The error of a UL+ algorithm will, at best, mirror the error of a UL algorithm with the correct sign (e.g. $\Theta\left(d/n_u\right)$ for the excess risk). Thus, despite using both labeled and unlabeled data, UL+ algorithms bear a close resemblance to UL algorithms that only use unlabeled data.

## 2.4 Brief overview of prior error bounds for SSL

In this section, we discuss prior theoretical works that aim to show benefits and limitations of SSL.

**Upper bounds** There are numerous known upper bounds on the excess risk of SSL algorithms for $\mathcal{P}_{\text{2-GMM}}$ distributions. However, despite showing better dependence on the labeled set size, these earlier bounds primarily match the UL+ rates [30, 31] or exhibit slower rates than UL+ [18]. That is, we cannot conclude from existing results that SSL algorithms can consistently outperform *both* SL and UL+, which is the question we aim to address in this paper.

**Lower bounds** In contrast to the upper bounds that aim to demonstrate benefits of SSL, three distinct minimax lower bounds for SSL have been proposed to show the limitations of SSL. Each proves, in different settings, that there exists a distribution $P_{XY}$ where SSL cannot outperform the SL minimax rate. Ben-David et al. [6] substantiate this claim for learning thresholds from univariate data

sourced from a uniform distribution on $[0, 1]$. Göpfert et al. [20] expand upon this by considering arbitrary marginal distributions $P_X$ and a "rich" set of realizable labeling functions, such that no volume of unlabeled data can differentiate between possible hypotheses. Lastly, Tolstikhin and Lopez-Paz [36] set a lower bound for scenarios with no implied association between the labeling function and the marginal distribution, a condition recognized as being unfavorable for SSL improvements [32]. Each of the aforementioned results contends that a particular worst-case distribution $P_{XY}$ exists, where the labeled sample complexity for SSL matches that of SL, even with limitless unlabeled data.

To summarize, the upper bounds show that SSL improves upon SL in settings where $n_u$ is significantly larger than $n_l$. In fact, for such large unlabeled sample size even UL+ can achieve small error. On the other hand, the lower bounds prove the futility of SSL for distributions where even infinite unlabeled data cannot help i.e. $P_X$ does not contain sufficient information about the conditional distribution. However these works fail to answer the question: does there exist a relatively moderate $n_u$ regime where SSL is better than both SL and UL+ ?

In the family of $\mathcal{P}_{\text{2-GMM}}$ distributions, the above lower bounds translate to the hard setting where $n_u \ll 1/s$. We now aim to prove a minimax lower bound for a fixed difficulty $s > 0$, that allows us to answer the above question in different regimes in terms of $n_l, n_u$.

## 3 Minimax rates for SSL

In this section we provide tight minimax lower bounds for SSL algorithms and 2-GMM distributions in $\mathcal{P}_{\text{2-GMM}}^{(s)}$. Our results indicate that it is, in fact, not possible for SSL algorithms to simultaneously achieve faster minimax rates than both SL and UL+.

### 3.1 Minimax rate

We begin by introducing tight lower bounds on the excess risk (4) of a linear estimator obtained using both labeled and unlabeled data. In addition, we also present tight lower bounds on estimation error (3) for the means of class-conditional distributions obtained similarly using labelled and unlabelled data. This is especially relevant when addressing linear classification of symmetric and spherical GMMs. In this setting, a reduced estimation error points to not only a low excess risk but also suggests a small calibration error under the assumption of a logistic noise model [29]. Both of these results are presented in Theorem 1. We present short proof sketches here and relegate the formal conditions required by the theorem to hold as well as the full proofs to Appendices B and C (for the estimation error and excess risk, respectively).

**Theorem 1** (SSL Minimax Rate for Excess Risk and Estimation Error). *Assume the conditions in Proposition 1 and additionally let $n_l > O(\frac{\log n_u}{s^2})$. Then for any $s \in (0, 1]$, we have*

$$\inf_{\mathcal{A}_{SSL}} \sup_{\|\boldsymbol{\theta}^*\|=s} \mathbb{E}\left[\mathcal{E}\left(\mathcal{A}_{SSL}\left(\mathcal{D}_l, \mathcal{D}_u\right), \boldsymbol{\theta}^*\right)\right] \asymp e^{-s^2/2} \min\left\{s, \frac{d}{sn_l + s^3 n_u}\right\}, \quad and$$

$$\inf_{\mathcal{A}_{SSL}} \sup_{\|\boldsymbol{\theta}^*\|=s} \mathbb{E}\left[\mathcal{R}_{estim}(\mathcal{A}_{SSL}(\mathcal{D}_l, \mathcal{D}_u), \boldsymbol{\theta}^*)\right] \asymp \min\left\{s, \sqrt{\frac{d}{n_l + s^2 n_u}}\right\},$$

*where the infimum is over all the possible SSL algorithms that have access to both unlabeled and labeled data and the expectation is over $\mathcal{D}_l \sim \left(P_{XY}^{\boldsymbol{\theta}^*}\right)^{n_l}$ and $\mathcal{D}_u \sim \left(P_X^{\boldsymbol{\theta}^*}\right)^{n_u}$.*

In the rest of the section, we focus solely on the bound for excess risk; however, we note that the discussion here transfers to estimation error as well. In Section 3.2, we discuss the new insights that this bound provides. A direct implication of the result is that $\epsilon_{\text{SSL}}\left(n_l, n_u, \Theta^{(s)}\right) \asymp \min\left(\epsilon_{\text{SL}}\left(n_l, 0, \Theta^{(s)}\right), \epsilon_{\text{UL+}}\left(n_l, n_u, \Theta^{(s)}\right)\right)$, that is, the minimax rate of SSL is the same as either that of SL or UL+, depending on the value of $s$ and the rate of growth of $n_u$ compared to $n_l$. Therefore, we can conclude that *no SSL algorithm can simultaneously improve the rates of both SL and UL+ for $\theta \in \Theta^{(s)}$.* We provide a full discussion of the rate improvements in more detail in Section 3.2.

**Proof sketch** The proof of the lower bound for excess risk is presented in Appendix C. For this proof, we adopt the packing construction in Li et al. [24] and apply Fano's method. Since the algorithms have access to both labeled and unlabeled datasets in the semi-supervised setting, KL

divergences between both the marginal and the joint distributions appear in the lower bound after the application of Fano's method, which is the key difference from its SL and UL counterparts.

We then show that the rate can be matched by the **SSL Switching Algorithm (SSL-S)** in Algorithm 2 – an oracle algorithm that switches between using a (minimax optimal) SL or UL+ algorithm based on the values of $s, n_l$, and $n_u$. The upper bound then follows as a corollary from Proposition 1 and the upper bounds for supervised learning.

For the parameter estimation error lower bound, we use Fano's method with the packing construction in Wu and Zhou [39], who have employed this method to derive lower bounds in the context of unsupervised learning. Similar to the excess risk analysis, the lower bound reveals that the SSL rate is either determined by the SL rate or the UL+ rate depending on $s$ and the ratio of the sizes of the labeled and unlabeled samples. Once again, the minimax error rate is matched by the SSL Switching algorithm presented in Algorithm 2.

---

**Algorithm 2:** SSL-S algorithm

**Input :** $\mathcal{D}_l, \mathcal{D}_u, s, \mathcal{A}_{\text{SL}}, \mathcal{A}_{\text{UL+}}$
**Result:** $\hat{\boldsymbol{\theta}}_{\text{SSL-S}}$
$\hat{\boldsymbol{\theta}}_{\text{SL}} \leftarrow \mathcal{A}_{\text{SL}}(\mathcal{D}_l)$
$\hat{\boldsymbol{\theta}}_{\text{UL+}} \leftarrow \mathcal{A}_{\text{UL+}}(\mathcal{D}_u, \mathcal{D}_l)$
**if** $s \leq \min\left\{\sqrt{\frac{d}{n_l}}, \left(\frac{d}{n_u}\right)^{1/4}\right\}$

    $\lfloor$   $\hat{\boldsymbol{\theta}}_{\text{SSL-S}} = \mathbf{0}$

**else if** $\min\left\{\sqrt{\frac{d}{n_l}}, \left(\frac{d}{n_u}\right)^{1/4}\right\} < s \leq \sqrt{\frac{n_l}{n_u}}$

    $\lfloor$   $\hat{\boldsymbol{\theta}}_{\text{SSL-S}} = \hat{\boldsymbol{\theta}}_{\text{SL}}$

**else**

    $\lfloor$   $\hat{\boldsymbol{\theta}}_{\text{SSL-S}} = \hat{\boldsymbol{\theta}}_{\text{UL+}}$

**return** $\hat{\boldsymbol{\theta}}_{\text{SSL-S}}$

---

**Discussion of the details of the theorem** We note that the SSL-S algorithm chooses to output the trivial estimator $\hat{\boldsymbol{\theta}}_{\text{SSL-S}} = \mathbf{0}$ if the SNR $s$ is low. In the low-SNR regime, the trivial estimator $\hat{\boldsymbol{\theta}} = \mathbf{0}$ achieves a small excess risk of $\mathcal{E}(\mathbf{0}, \boldsymbol{\theta}^*) = \Phi(s) - \Phi(0) \simeq e^{-s^2/2}s$, where $\Phi$ is the CDF of the standard normal distribution.

Moreover, the technical condition $s \in (0, 1]$ is not overly restrictive as this range for the SNR is already sufficient for seeing all the relevant regimes, e.g. SSL-S switches from using SL to UL+ within this range. Finally, while the *rates* of either SL or UL+ cannot be improved further using SSL algorithms, it is nonetheless possible to improve the error by a constant factor, independent of $n_l$ and $n_u$. To see this, in Section 4 we describe an algorithm that uses both $\mathcal{D}_l$ and $\mathcal{D}_u$ effectively and can hence achieve a provable improvement in error over both SL and UL+.

### 3.2 Comparison of SSL with UL+ and SL in different regimes

To understand whether an SSL algorithm is using the labeled and unlabeled data effectively, we compare the error rate of SSL algorithms to the minimax rates for SL and UL+ algorithms.

A gap in these rates for a certain SSL algorithm would indicate that the algorithm can obtain lower error than both SL and UL+ when presented with the same amount of (large enough) data. We study the *improvement rates* defined below, which capture the error ratio for the worst-case distributions between the minimax rate for SSL and the minimax rate for SL or UL+.

**Definition 1** (SSL improvement rates). *For a set of parameters $\Theta \subseteq \mathbb{R}^d$, we define the improvement rates of SSL over SL and UL+ as $h_l$ and $h_u$, respectively, where*

$$h_l(n_l, n_u, \Theta) := \frac{\inf_{\mathcal{A}_{SSL}} \sup_{\boldsymbol{\theta}^* \in \Theta} \mathbb{E}\left[\mathcal{E}\left(\mathcal{A}_{SSL}(\mathcal{D}_l, \mathcal{D}_u), \boldsymbol{\theta}^*\right)\right]}{\inf_{\mathcal{A}_{SL}} \sup_{\boldsymbol{\theta}^* \in \Theta} \mathbb{E}\left[\mathcal{E}\left(\mathcal{A}_{SL}(\mathcal{D}_l, \emptyset), \boldsymbol{\theta}^*\right)\right]}, \tag{6}$$

$$h_u(n_l, n_u, \Theta) := \frac{\inf_{\mathcal{A}_{SSL}} \sup_{\boldsymbol{\theta}^* \in \Theta} \mathbb{E}\left[\mathcal{E}\left(\mathcal{A}_{SSL}(\mathcal{D}_l, \mathcal{D}_u), \boldsymbol{\theta}^*\right)\right]}{\inf_{\mathcal{A}_{UL+}} \sup_{\boldsymbol{\theta}^* \in \Theta} \mathbb{E}\left[\mathcal{E}\left(\mathcal{A}_{UL+}(\mathcal{D}_l, \mathcal{D}_u), \boldsymbol{\theta}^*\right)\right]}, \tag{7}$$

*where the expectations are over $\mathcal{D}_l \sim \left(P_{XY}^{\boldsymbol{\theta}^*}\right)^{n_l}$ and $\mathcal{D}_u \sim \left(P_X^{\boldsymbol{\theta}^*}\right)^{n_u}$.*

To simplify notation, we denote the improvement rates of SL and UL+ over $\Theta^{(s)}$ as $h_l(n_l, n_u, s)$ and $h_u(n_l, n_u, s)$, respectively. A straightforward upper bound for these rates is $h_l, h_u \leq 1$, achieved when utilizing an SL and UL+ algorithm, respectively. SSL demonstrates an enhanced error rate over SL and UL+, if both $\lim_{n_l, n_u \to \infty} h_l(n_l, n_u, \Theta) = 0$ and $\lim_{n_l, n_u \to \infty} h_u(n_l, n_u, \Theta) = 0$. If $h_l$ or $h_u$ lies in $(0, 1)$ without converging to zero as $n_l, n_u \to \infty$, then SSL surpasses SL or UL+, respectively, only by a constant factor.

| SNR Regime | Rate of growth of $n_u$ vs $n_l$ | $h_l(n_l, n_u, s)$ | $h_u(n_l, n_u, s)$ |
|---|---|---|---|
| $s = o\left(\sqrt{1/n_u}\right)$ | Any | $c_{\text{SL}}$ | $0$ |
| fixed $s > 0$ | $n_u = o\left(n_l\right)$ | $c_{\text{SL}}$ | $0$ |
| | $n_u = \omega(n_l)$ | $0$ | $c_{\text{UL}}$ |
| | $\lim_{n_l, n_u \to \infty} \frac{n_u}{n_l} = c$ | $\left(\frac{1}{1+cs^2}\right) c_{\text{SL}}$ | $\left(\frac{s^2 c}{1+s^2 c}\right) c_{\text{UL}}$ |

Table 2: SSL improvement rates over SL and UL+ for different regimes of $s, n_l$ and $n_u$, where $h_l$ and $h_u$ are evaluated for $\lim_{n_l, n_u \to \infty}$.

Given Theorem 1, we can directly derive the improvement rates of SSL in the following corollary.

**Corollary 1.** *Assuming the setting of Theorem 1, the improvement rates of SSL can be written as:*

$$\textit{Improvement rate over SL: } h_l\left(n_l, n_u, s\right) \asymp \frac{n_l}{n_l + s^2 n_u}.$$

$$\textit{Improvement rate over UL+: } h_u\left(n_l, n_u, s\right) \asymp \frac{s^2 n_u}{n_l + s^2 n_u}.$$

Based on this corollary, we now argue how no SSL algorithm can simultaneously improve the rates of both SL and UL+ for any regime of $s, n_l$ and $n_u$, as summarized in Table 2.

**1. SSL rate is not faster than SL, but is faster than UL+ .** For extremely low values of SNR (i.e. $s \to 0$ faster than $\sqrt{n_u} \to \infty$), we have that $\lim_{n_l, n_u \to \infty} h_l(n_l, n_u, s) = c_{SL} > 0$ and hence SSL fails to improve the SL rates even with infinite unlabeled data. This setting has been the focus of previous worst-case analyses for SSL [6, 36, 20]. Alternatively, the SSL rate is also not better than the SL rate for a fixed SNR, when the labeled dataset is significantly larger than the unlabeled data, i.e. $n_u = o(n_l)$.

**2. SSL rate is faster than SL, but not faster than UL+.** As mentioned in Göpfert et al. [20], in order for SSL to lead to a rate improvement compared to SL, it is *necessary* that the unlabeled set size is at least a superlinear function of the labeled set size, i.e. $n_u = \omega(n_l)$. Corollary 1 shows that this condition is, in fact, *sufficient* for 2-GMM distributions: for $s > 0$, as long as $n_u$ grows superlinearly with respect to $n_l$, $h_l(n_l, n_u, s) \to 0$, and hence, SSL can achieve faster rates than SL. Despite the improvement over SL, for this setting, the asymptotic error ratio between SSL and UL+ does not vanish, i.e. $\lim_{n_l, n_u \to \infty} h_u\left(n_l, n_u, s\right) = c_{\text{UL}} > 0$.

**3. SSL rate is not faster than either SL or UL+.** Finally, in the regime where the unlabeled dataset size depends linearly on the size of the labeled set i.e. $\lim_{n_l, n_u \to \infty} \frac{n_u}{n_l} \to c$ for some constant $c > 0$, neither of the improvement rates vanishes for $n_l, n_u \to \infty$.

As explained in Section 2.2, prior UL+ algorithms that work well in practice do not use the labeled data for the unsupervised learning step. Interestingly, for the case when both the labeled and unlabeled data are used for the unsupervised step, the trends in Table 2 remain the same, with only one exception: the improvement rates in the last line of the table (i.e. $\frac{n_u}{n_l} \to c \in (0, \infty)$) would only change by a small constant. More importantly, the main takeaway from Table 2 remains the same: SSL cannot achieve better rates than both UL+ and SL at the same time since there is no regime for which $h_l$ and $h_u$ are simultaneously 0.

## 4 Empirical improvements over the rate-optimal SSL-S algorithm

Section 3 shows that the SSL minimax rates can be achieved with a simple algorithm that switches between a minimax optimal SL and a minimax optimal UL+ algorithm. Despite being optimal in terms of statistical rates, this SSL Switching algorithm does not make the most effective use of the available data: SL algorithms solely uses the labeled data whereas UL+ learns the decision boundary using just the unlabeled data and the labeled data is used to only label the different prediction regions. Alternatively, one could use both labeled and unlabeled data to learn the decision boundary. In this section, we investigate the following question: Is it possible to improve over the error of SSL-S even though the rate would be, at best, as good as the 'wasteful' SSL-S algorithm as per Section 3?

In this section, we present experiments to show that, in fact, a remarkably simple algorithm (i.e. a weighted ensemble of the SL and UL+ classifiers) can outperform the minimax optimal SSL-S algorithm. We show that algorithms such as self-training, which have been shown to excel in practice [40, 34, 22], can also improve over SSL-S. Consequently, it remains an exciting avenue for future work to derive tight analyses that characterize the improvement of such algorithms over SL/UL+.

**A simple algorithm more effective than SSL-S** A natural approach to improve upon the naive SSL-S algorithm is to construct a weighted ensemble of an SL and a UL+ estimator, trained on $\mathcal{D}_l$ and $\mathcal{D}_u$, respectively, with a controllable weighting hyperparameter $t$. We call this the **SSL Weighted algorithm (SSL-W)** shown in Algorithm 3. With an appropriate choice of $t$, it is possible to show that the SSL-W algorithm performs better (up to sign permutation) than SSL-S. The formal statement of this re-

---

**Algorithm 3:** SSL-W algorithm

**Input :** $\mathcal{D}_l, \mathcal{D}_u, t$
**Result:** $\hat{\boldsymbol{\theta}}_{\text{SSL-W}}$
$\hat{\boldsymbol{\theta}}_{\text{SL}} \leftarrow \mathcal{A}_{\text{SL}}(\mathcal{D}_l)$
$\hat{\boldsymbol{\theta}}_{\text{UL+}} \leftarrow \mathcal{A}_{\text{UL+}}(\mathcal{D}_l, \mathcal{D}_u)$
$\hat{\boldsymbol{\theta}}_{\text{SSL-W}}(t) = t\hat{\boldsymbol{\theta}}_{\text{SL}} + (1-t)\hat{\boldsymbol{\theta}}_{\text{UL+}}$
**return** $\hat{\boldsymbol{\theta}}_{\text{SSL-W}}(t)$

---

sult together with the proof are deferred to Appendix D. In practice, one can fix the sign permutation of the $\hat{\boldsymbol{\theta}}_{\text{SSL-W}}$ estimator using a small amount of labeled data. The intuition for this improvement is that the ensemble estimator $\hat{\boldsymbol{\theta}}_{\text{SSL-W}}$ achieves lower error than the constituent estimators of the ensemble (i.e. $\hat{\boldsymbol{\theta}}_{\text{SL}}$ and $\hat{\boldsymbol{\theta}}_{\text{UL+}}$), which, in turn, determine the error of the SSL-S algorithm.

## 4.1 Empirical improvements over the minimax optimal SSL-S algorithm

In this section we present linear classification experiments on synthetic and real-world data to show that there indeed exist SSL algorithms that can improve over the error of the (minimax optimal) SSL Switching Algorithm. For both synthetic and real-world data, we use $\hat{\boldsymbol{\theta}}_{\text{SL}} = \frac{1}{n_l} \sum_{i=1}^{n_l} Y_i X_i$ as the SL estimator and an Expectation-Maximization (EM) algorithm for the UL algorithm (see Appendix E for details). The optimal switching point for SSL-S and the optimal weight for SSL-W, as well as the optimal $\ell_2$ penalty for logistic regression are chosen using a validation set.

**Synthetic data.** We consider data drawn from a symmetric and isotropic 2-GMM distribution $P_{XY}^{\boldsymbol{\theta}^*}$ over $\mathbb{R}^2$. The labeled and unlabeled set sizes are set to 20 and 2000, respectively. Figure 1a shows the gap between SSL-W and SL or UL+ as a function of the SNR $s$ (Figure 3 in Appendix F shows the dependence of the error gap on $n_l$). There are two main takeaways. First, for varying SNR values $s$, SSL-W always outperforms SL and UL+, and hence, also SSL-S. Second, as argued in Section 3.2, SSL-W improves more over UL+ for small values of the SNR $s$, and it improves more over SL for large values of the SNR.

**Real-world data.** We consider 10 binary classification real-world datasets: five from the OpenML repository [37] and five 2-class subsets of the MNIST dataset [13]. For the MNIST subsets, we choose class pairs that have a linear Bayes error varying between $0.1\%$ and $2.5\%$.[4] We choose from OpenML datasets that have a large enough number of samples compared to dimensionality (see Appendix E for details on how we choose the datasets). The OpenML datasets span a range of Bayes errors that varies between $3\%$ and $34\%$.

In the absence of the exact data-generating process, we quantify the difficulty of SSL on real-world datasets using a notion of *compatibility*, reminiscent of Balcan and Blum [4]. Specifically, we consider the compatibility given by $\rho^{-1}$ with $\rho := \frac{1}{2\sqrt{d}} \left( \Delta(\boldsymbol{\theta}_{UL+}^*, \boldsymbol{\theta}_{\text{Bayes}}^*) + \mathcal{R}_{\text{pred}}(\boldsymbol{\theta}_{\text{Bayes}}^*) \right)$, where $\Delta(\boldsymbol{\theta}_{UL+}^*, \boldsymbol{\theta}_{\text{Bayes}}^*) := \frac{\mathcal{R}_{\text{pred}}(\boldsymbol{\theta}_{UL+}^*) - \mathcal{R}_{\text{pred}}(\boldsymbol{\theta}_{\text{Bayes}}^*)}{\mathcal{R}_{\text{pred}}(\boldsymbol{\theta}_{\text{Bayes}}^*)}$, $d$ is the dimension of the data, $\boldsymbol{\theta}_{\text{Bayes}}^*$ is obtained via SL on the entire dataset and $\boldsymbol{\theta}_{\text{UL+}}^*$ determines the predictor with optimal sign obtained via UL on the entire dataset. Intuitively, this notion of compatibility captures both the Bayes error of a dataset, as well as how compatible the 2-GMM parametric assumption actually is for the given data.

In addition to SSL-S (Algorithm 2) and SSL-W (Algorithm 3) we also evaluate the performance of self-training, using a procedure similar to the one analyzed in Frei et al. [18]. We use a logistic regression estimator for the pseudolabeling, and train logistic regression with a ridge penalty in the second stage of the self-training procedure. Note that an $\ell_2$ penalty corresponds to input consistency regularization [38] with respect to $\ell_2$ perturbations.

---

[4]We estimate the Bayes error of a dataset by training a linear classifier on the entire labeled dataset.

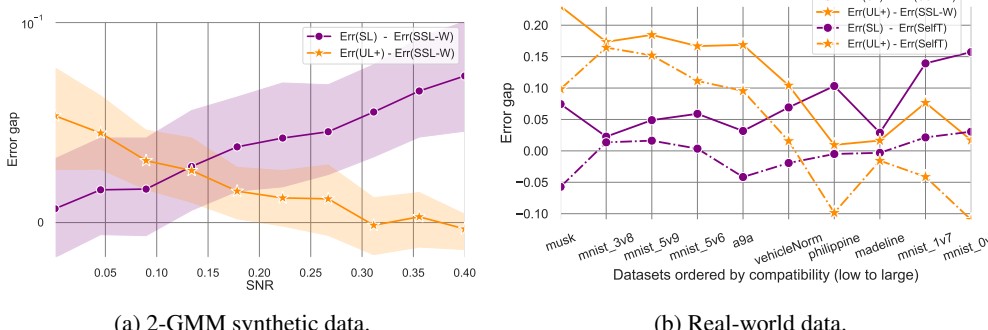

(a) 2-GMM synthetic data.

(b) Real-world data.

Figure 1: Error gap between SL or UL+ and SSL-W for varying task difficulties. We assess problem difficulty using the SNR (for 2-GMMs) or the compatibility $\rho^{-1}$ (for real-world data). We see the same trends for both synthetic and real-world data. Moreover, self-training also exhibits the same trend as $\hat{\boldsymbol{\theta}}_{\text{SSL-W}}$.

Figure 1b shows the improvement in classification error of SSL algorithms (i.e. SSL-W and self-training) compared to SL and UL+ . The positive values of the error gap indicate that SSL-W outperforms both SL and UL+ even on real-world datasets. This finding suggests that the intuition presented in this section carries over to more generic distributions beyond just 2-GMMs. Finally, Figure 4 in Appendix F shows that SSL-W improves over the error of the minimax optimal SSL-S algorithm, for varying values of $n_l$. These findings provide some proof-of-concept evidence that existing SSL algorithms may already be obtaining lower error than the minimax optimal SSL-S algorithm. We hope this observation will encourage future research into characterizing this error gap for practically relevant algorithms such as self-training.

## 5    Related work

**Other theoretical analyses of SSL algorithms.**    Beyond the theoretical studies highlighted in Section 2, there are a few others pertinent to our research. Specifically, Azizyan et al. [1], Singh et al. [33] present upper bounds for semi-supervised regression, which are contingent on the degree to which the marginal $P_X$ informs the labeling function. This is akin to the results we derive in this work. However, obtaining a minimax lower bound for semi-supervised regression remains an exciting direction for future work. We refer to [27] for an overview of prior theoretical results for SSL.

Balcan and Blum [4] introduced a compatibility score, denoted as $\chi(f, P_X) \in [0, 1]$, which connects the space of marginal distributions to the space of labeling functions. While their findings hint that SSL may surpass the SL minimax rates, they offer no comparisons with UL/UL+. Moreover, the paper does not discuss minimax optimality of the proposed SSL algorithms.

On another note, even though SSL does not enhance the rates of UL, Sula and Zheng [35] demonstrate that labeled samples can bolster the convergence speed of Expectation-Maximization within the context of our study.

To conclude, Schölkopf et al. [32] leveraged a causality framework to pinpoint scenarios where SSL does not offer any advantage over SL. In essence, when the covariates, represented by $X$, act as causal ancestors to the labels $Y$, the independent causal mechanism assumption dictates that the marginal $P_X$ offers no insights about the labeling function.

**Minimax rates for SL and UL.**    The proofs in this work rely on techniques used to derive minimax rates for SL and UL algorithms. Most of these prior results consider the same distributional assumptions as our paper. Wu and Zhou [39] show a tight minimax lower bound for estimation error for spherical 2-GMMs from $\mathcal{P}_{\text{2-GMM}}$. Moreover, Azizyan et al. [2], Li et al. [24] derive minimax rates over $\mathcal{P}_{\text{2-GMM}}$ for classification and clustering (up to permutation).

In addition to the SL and UL algorithms considered in Section 3, Expectation-Maximization (EM) is another family of algorithms that is commonly analyzed for the same distributional setting considered in our paper. For instance, Wu and Zhou [39] rely on techniques from several previous seminal papers [12, 3, 14–16] to obtain upper bounds for EM-style algorithms.

## 6   Conclusions and limitations

In this paper, we establish a tight lower bound for semi-supervised classification within the class of 2-GMM distributions. Our findings demonstrate that SSL cannot simultaneously improve the error rates of both SL and UL across all signal-to-noise ratios. However, empirical evidence suggests that SSL *can* improve upon the error of minimax optimal SL or UL algorithms. This observation calls for careful analyses of the error of SSL algorithms that also track constant factors, not only rates.

Our theoretical analysis focuses exclusively on isotropic and symmetric GMMs due to limitations in the technical tools employed for the proofs. Similar constraints can be observed in recent analyses of SL or UL algorithms [24, 39]. However, it is worth noting that whenever the bounds for SL and UL can be extended to more general distributions in the future, these results can be seamlessly used to also extend Theorem 1 to these settings.

## Acknowledgement

AT was supported by a PhD fellowship from the Swiss Data Science Center. We also thank the anonymous reviewers for their helpful comments.

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

# A Proof of Proposition 1

In this section we provide the formal statement of Proposition 1 and its proof.

**Proposition 2** (Fixing the sign of $\hat{\boldsymbol{\theta}}_{\mathrm{UL}}$)**.** *There exist universal constants $c_0, C_1, C_2, C_3, C_4 > 0$ such that for $n_u \geq (160/s)^2 d$, $d \geq 2$ and $s \in (0,1]$, it holds that:*

$$\mathbb{E}\left[\|\hat{\boldsymbol{\theta}}_{UL+} - \boldsymbol{\theta}^*\|\right] \leq C_1 \sqrt{\frac{d}{s^2 n_u}} + C_2 s e^{-\frac{1}{2} n_l s^2 (1 - c_0 \sqrt{\frac{d \log(n_u)}{s^2 n_u}})^2}, \quad and$$

$$\mathbb{E}\left[\mathcal{E}(\hat{\boldsymbol{\theta}}_{UL+})\right] \leq C_3 e^{-\frac{1}{2} s^2} \frac{d \log(dn_u)}{s^3 n_u} + C_4 e^{-\frac{1}{2} s^2 n_l \left(1 - c_0 \sqrt{\frac{d \log(n_u)}{s^2 n_u}}\right)^2}.$$

*Proof.* Recall that we consider the UL+ estimator $\hat{\boldsymbol{\theta}}_{\mathrm{UL}+} = \mathrm{sign}\left(\hat{\boldsymbol{\theta}}_{\mathrm{SL}}^\top \hat{\boldsymbol{\theta}}_{\mathrm{UL}}\right) \hat{\boldsymbol{\theta}}_{\mathrm{UL}}$ and denote $\hat{\beta} := \mathrm{sign}\left(\hat{\boldsymbol{\theta}}_{\mathrm{SL}}^\top \hat{\boldsymbol{\theta}}_{\mathrm{UL}}\right)$. Now let $\beta := \mathrm{sign}(\boldsymbol{\theta}^{*\top} \hat{\boldsymbol{\theta}}_{\mathrm{UL}}) = \arg\min_{\tilde{\beta} \in \{-1,+1\}} \|\tilde{\beta} \hat{\boldsymbol{\theta}}_{\mathrm{UL}} - \boldsymbol{\theta}^*\|^2$.

Note that we can write the expected estimation error of $\hat{\boldsymbol{\theta}}_{\mathrm{UL}+}$ as

$$
\begin{aligned}
\mathbb{E}\left[\|\hat{\boldsymbol{\theta}}_{\mathrm{UL}+} - \boldsymbol{\theta}^*\|\right] &= \mathbb{E}\left[\|\hat{\beta} \hat{\boldsymbol{\theta}}_{\mathrm{UL}} - \boldsymbol{\theta}^*\|\right] \\
&= \mathbb{E}\left[\mathbb{1}_{\{\hat{\beta}=\beta\}} \|\beta \hat{\boldsymbol{\theta}}_{\mathrm{UL}} - \boldsymbol{\theta}^*\| + \mathbb{1}_{\{\hat{\beta} \neq \beta\}} \|\beta \hat{\boldsymbol{\theta}}_{\mathrm{UL}} + \boldsymbol{\theta}^*\|\right] \\
&\leq \mathbb{E}\left[\mathbb{1}_{\{\hat{\beta}=\beta\}} \|\beta \hat{\boldsymbol{\theta}}_{\mathrm{UL}} - \boldsymbol{\theta}^*\|\right] + \mathbb{E}\left[\mathbb{1}_{\{\hat{\beta} \neq \beta\}} (\|\beta \hat{\boldsymbol{\theta}}_{\mathrm{UL}} - \boldsymbol{\theta}^*\| + 2\|\boldsymbol{\theta}^*\|)\right] \\
&\leq \mathbb{E}\left[\|\beta \hat{\boldsymbol{\theta}}_{\mathrm{UL}} - \boldsymbol{\theta}^*\|\right] + 2s \mathbb{P}(\hat{\beta} \neq \beta).
\end{aligned}
\tag{8}
$$

Next, we recall a result by Wu and Zhou [39] established for this particular UL estimator, namely that $\mathbb{E}\left[\|\beta \hat{\boldsymbol{\theta}}_{\mathrm{UL}} - \boldsymbol{\theta}^*\|\right] \lesssim \sqrt{\frac{d}{s^2 n_u}}$. Moreover, the probability of incorrectly estimating the sign (permutation) can be written as

$$
\begin{aligned}
\mathbb{P}(\hat{\beta} \neq \beta) &= \mathbb{P}\left(\mathrm{sign}\left(\hat{\boldsymbol{\theta}}_{\mathrm{SL}}^\top \hat{\boldsymbol{\theta}}_{\mathrm{UL}}\right) \neq \mathrm{sign}\left(\boldsymbol{\theta}^{*\top} \hat{\boldsymbol{\theta}}_{\mathrm{UL}}\right)\right), \text{ where } \hat{\boldsymbol{\theta}}_{\mathrm{SL}} \sim \mathcal{N}(\boldsymbol{\theta}^*, \frac{1}{n_l} I_d) \\
&\leq \mathbb{P}\left(\mathrm{sign}(\tilde{Z}) \neq \mathrm{sign}\left(\boldsymbol{\theta}^{*\top} \hat{\boldsymbol{\theta}}_{\mathrm{UL}}\right)\right), \text{ where } \tilde{Z} \sim \mathcal{N}(\hat{\boldsymbol{\theta}}_{\mathrm{UL}}^\top \boldsymbol{\theta}^*, \frac{1}{n_l}(\hat{\boldsymbol{\theta}}_{\mathrm{UL}}^\top \hat{\boldsymbol{\theta}}_{\mathrm{UL}})) \\
&\leq \mathbb{P}\left(Z' \geq |\hat{\boldsymbol{\theta}}_{\mathrm{UL}}^\top \boldsymbol{\theta}^*|\right), \text{ where } Z' \sim \mathcal{N}(0, \frac{1}{n_l}(\hat{\boldsymbol{\theta}}_{\mathrm{UL}}^\top \hat{\boldsymbol{\theta}}_{\mathrm{UL}})) \\
&= \mathbb{P}\left(Z \geq \sqrt{n_l s^2} S_C(\hat{\boldsymbol{\theta}}_{\mathrm{UL}}, \boldsymbol{\theta}^*)\right),
\end{aligned}
$$

where $Z \sim \mathcal{N}(0,1)$ and $S_C(\hat{\boldsymbol{\theta}}_{\mathrm{UL}}, \boldsymbol{\theta}^*) = \frac{|\hat{\boldsymbol{\theta}}_{\mathrm{UL}}^\top \boldsymbol{\theta}^*|}{\|\hat{\boldsymbol{\theta}}_{\mathrm{UL}}\| \|\boldsymbol{\theta}^*\|}$ Therefore, for any $A$ we have:

$$
\begin{aligned}
\mathbb{P}(\hat{\beta} \neq \beta) &\leq \mathbb{P}(Z \geq \sqrt{n_l s^2}(1 - A)) + \mathbb{P}\left(S_C(\hat{\boldsymbol{\theta}}_{\mathrm{UL}}, \boldsymbol{\theta}^*) \leq 1 - A\right) \\
&\leq e^{-\frac{1}{2} n_l s^2 (1-A)^2} + \mathbb{P}\left(S_C(\hat{\boldsymbol{\theta}}_{\mathrm{UL}}, \boldsymbol{\theta}^*) \leq 1 - A\right),
\end{aligned}
$$

where we used the Chernoff bound in the last step. Finally, setting $A = c_0 \sqrt{\frac{d \log(n_u)}{s^2 n_u}}$ as a corollary of Proposition 6 in Azizyan et al. [2] for $n_u \geq (160/s)^2 d$ we have $\mathbb{P}\left(S_C(\hat{\boldsymbol{\theta}}_{\mathrm{UL}}, \boldsymbol{\theta}^*) \leq 1 - A\right) \leq \frac{d}{n_u}$. Therefore, for $n_u \geq (160/s)^2 d$, we have the following upper bound on estimating the sign incorrectly:

$$\mathbb{P}(\hat{\beta} \neq \beta) \leq e^{-\frac{1}{2} n_l s^2 \left(1 - c_0 \sqrt{\frac{d \log(n_u)}{s^2 n_u}}\right)^2} + \frac{d}{n_u}. \tag{9}$$

Combining this result with Equation (8) finishes the proof of the estimation error bound in the proposition, as we obtain that for some $C_1, C_2 > 0$, the following holds:

$$\mathbb{E}\left[\|\hat{\boldsymbol{\theta}}_{\text{UL+}} - \boldsymbol{\theta}^*\|\right] \leq C_1 \sqrt{\frac{d}{s^2 n_u}} + C_2 s e^{-\frac{1}{4} n_l s^2 \left(1 - c_0 \sqrt{\frac{d \log(n_u)}{s^2 n_u}}\right)^2}.$$

Similarly, we can decompose the expected excess risk as follows:

$$
\begin{aligned}
\mathbb{E}\left[\mathcal{E}(\hat{\boldsymbol{\theta}}_{\text{UL+}})\right] = \mathbb{E}\left[\mathcal{E}(\hat{\beta}\hat{\boldsymbol{\theta}}_{\text{UL}})\right] &= \mathbb{E}\left[\mathbb{1}_{\{\hat{\beta}=\beta\}}\mathcal{E}(\hat{\beta}\hat{\boldsymbol{\theta}}_{\text{UL}}) + \mathbb{1}_{\{\hat{\beta}\neq\beta\}}\mathcal{E}(\hat{\beta}\hat{\boldsymbol{\theta}}_{\text{UL}})\right] \\
&= \mathbb{E}\left[\mathbb{1}_{\{\hat{\beta}=\beta\}}\mathcal{E}(\beta\hat{\boldsymbol{\theta}}_{\text{UL}}) + \mathbb{1}_{\{\hat{\beta}\neq\beta\}}\mathcal{E}(-\beta\hat{\boldsymbol{\theta}}_{\text{UL}})\right] \\
&\leq \mathbb{E}\left[\mathcal{E}(\beta\hat{\boldsymbol{\theta}}_{\text{UL}}) + \mathbb{1}_{\{\hat{\beta}\neq\beta\}}\right] \\
&= \mathbb{E}\left[\mathcal{E}(\beta\hat{\boldsymbol{\theta}}_{\text{UL}})\right] + \mathbb{P}(\hat{\beta}\neq\beta),
\end{aligned}
\tag{10}
$$

where in Equation (10) we upper bound the indicator function and the excess risk terms by 1. Finally, combining Equation (9) with the upper bound in Li et al. [24] for $\mathbb{E}\left[\mathcal{E}(\beta\hat{\boldsymbol{\theta}}_{\text{UL}})\right]$, we obtain the desired result. More concretely, we get that for some $C_3, C_4 > 0$,

$$\mathbb{E}\left[\mathcal{E}(\hat{\boldsymbol{\theta}}_{\text{UL+}})\right] \leq C_3 e^{-\frac{1}{2} s^2} \frac{d \log(d n_u)}{s^3 n_u} + C_4 e^{-\frac{1}{2} s^2 n_l \left(1 - c_0 \sqrt{\frac{d \log(n_u)}{s^2 n_u}}\right)^2}.$$

$\square$

## B  Proof of SSL Minimax Rate for Estimation Error in Theorem 1

In this section we provide the proofs for the lower and upper bounds on the estimation error presented in Theorem 1. The proof for the excess risk rates can be found in Appendix C.

### B.1  Proof of estimation error lower bound

We first prove the estimation error lower bound in Theorem 1. As discussed in Section 2, consider the 2-GMM distributions from $\mathcal{P}_{\text{2-GMM}}^{(s)}$, with isotropic components and identical covariance matrices.

Consider an arbitrary set of predictors $\mathcal{M} = \{\boldsymbol{\theta}_i\}_{i=0}^{M}$. . We can apply Fano's method [11] to obtain that the following holds:

$$
\begin{aligned}
&\inf_{\mathcal{A}_{\text{SSL}}} \sup_{\|\boldsymbol{\theta}^*\|=s} \mathbb{E}_{\mathcal{D}_l, \mathcal{D}_u}\left[\mathcal{R}_{\text{estim}}(\mathcal{A}_{\text{SSL}}(\mathcal{D}_l, \mathcal{D}_u), P_{XY}^{\boldsymbol{\theta}^*})\right] \\
&\geq \frac{1}{2} \min_{\substack{i,j\in[M]\\i\neq j}} \|\boldsymbol{\theta}_i - \boldsymbol{\theta}_j\| \left(1 - \frac{1 + \frac{1}{M}\sum_{i=1}^{M} D\left(P_{XY}^{\boldsymbol{\theta}_i\ n_l} P_X^{\boldsymbol{\theta}_i\ n_u} \| P_{XY}^{\boldsymbol{\theta}_0\ n_l} P_X^{\boldsymbol{\theta}_0\ n_u}\right)}{log(M)}\right) \\
&= \frac{1}{2} \min_{\substack{i,j\in[M]\\i\neq j}} \|\boldsymbol{\theta}_i - \boldsymbol{\theta}_j\| \left(1 - \frac{1 + \frac{1}{M}\sum_{i=1}^{M} n_l D\left(P_{XY}^{\boldsymbol{\theta}_i} \| P_{XY}^{\boldsymbol{\theta}_0}\right) + n_u D\left(P_X^{\boldsymbol{\theta}_i} \| P_X^{\boldsymbol{\theta}_0}\right)}{log(M)}\right) \\
&\geq \frac{1}{2} \min_{\substack{i,j\in[M]\\i\neq j}} \|\boldsymbol{\theta}_i - \boldsymbol{\theta}_j\| \left(1 - \frac{1 + n_l \max_{i\in[M]} D\left(P_{XY}^{\boldsymbol{\theta}_i} \| P_{XY}^{\boldsymbol{\theta}_0}\right) + n_u \max_{i\in[M]} D\left(P_X^{\boldsymbol{\theta}_i} \| P_X^{\boldsymbol{\theta}_0}\right)}{log(M)}\right),
\end{aligned}
\tag{11}
$$

(12)

where $D\left(\cdot||\cdot\right)$ denotes the KL divergence. In Equation (11), we use the fact that the labeled and unlabeled samples are drawn i.i.d. from $P_X$ and $P_{XY}$ and in Equation (12) we upper bound the average with the maximum. The next step of the proof consists in choosing an appropriate packing $\{\boldsymbol{\theta}_i\}_{i=1}^M$ and $\boldsymbol{\theta}_0$ on the sphere of radius $s$, i.e. $\frac{1}{s}\boldsymbol{\theta}_i \in S^{d-1}$, that optimizes the trade-off between the minimum and the maxima in Equation (12).

For the packing, we use the same construction that was employed by Wu and Zhou [39] for deriving adaptive bounds for unsupervised learning. This construction has the advantage that it also leads to a tight lower bound for the supervised setting. Let $c_0$ and $C_0$ be positive absolute constants and let $\tilde{\mathcal{M}} = \{\psi_1, ..., \psi_M\}$ be a $c_0$-net on the unit sphere $S^{d-2}$ such that we have $|\tilde{\mathcal{M}}| = M \geq e^{C_0 d}$. For an absolute constant $\alpha \in [0, 1]$, we construct the following packing of the sphere of radius $s$ in $\mathbb{R}^d$:

$$\mathcal{M} = \left\{\boldsymbol{\theta}_i = s \begin{bmatrix} \sqrt{1-\alpha^2} \\ \alpha\psi_i \end{bmatrix} \middle| \psi_i \in \tilde{\mathcal{M}} \right\},$$

and define $\boldsymbol{\theta}_0 = [s, 0, ..., 0]$. Note that, by definition, $\|\boldsymbol{\theta}_i - \boldsymbol{\theta}_j\| \geq c_0 s\alpha$, for any distinct $i, j \in [M]$, which lower bounds the first term in (12). Furthermore, $\|\boldsymbol{\theta}_i - \boldsymbol{\theta}_0\| \leq \sqrt{2}\alpha s$, for all $i \in [M]$.

In the next step, we upper bound the maxima in Equation (12). First, we write the KL divergence between two GMMs with identity covariance matrices:

$$D\left(P_{XY}^{\boldsymbol{\theta}_i}||P_{XY}^{\boldsymbol{\theta}_0}\right) = \frac{1}{2}\|\boldsymbol{\theta}_i - \boldsymbol{\theta}_0\|_2^2 \leq \alpha^2 s^2, \text{ for all } i = [M]. \tag{13}$$

Second, we can upper bound the KL divergence between marginal distributions, namely $D\left(P_X^{\boldsymbol{\theta}_i}||P_X^{\boldsymbol{\theta}_0}\right)$, using Lemma 27 in Wu and Zhou [39], which implies that:

$$\max_{i\in[M]} D\left(P_X^{\boldsymbol{\theta}_i}||P_X^{\boldsymbol{\theta}_0}\right) \leq C \max_{i\in[M]}\|\frac{1}{s}\boldsymbol{\theta}_i - \frac{1}{s}\boldsymbol{\theta}_0\|^2 s^4 \leq 2C\alpha^2 s^4. \tag{14}$$

Plugging Equations (13) and (14) into Equation (12) we obtain the following lower bound for the minimax error, which holds for any $\alpha \leq 1$:

$$\inf_{\mathcal{A}_{\text{SSL}}} \sup_{\|\boldsymbol{\theta}^*\|=s} \mathbb{E}_{\mathcal{D}_l,\mathcal{D}_u}\left[\mathcal{R}_{\text{estim}}(\mathcal{A}_{\text{SSL}}(\mathcal{D}_l,\mathcal{D}_u),\boldsymbol{\theta}^*)\right] \geq \frac{1}{2}c_o\alpha s\left(1 - \frac{1 + n_l s^2\alpha^2 + n_u C_1 s^4\alpha^2}{C_0 d}\right).$$

Minimizing over $\alpha$ yields the optimum value $\alpha = \min\left\{1, \sqrt{\frac{C_0 d-1}{3s^2 n_l + 3C_1 s^4 n_u}}\right\}$, where the minimum comes from how we have constructed the packing, which requires that $\alpha \leq 1$. Using this value for $\alpha$ concludes the proof.

## B.2 Proof of estimation error upper bound

We now prove the tightness of our lower bound by establishing the upper bound for the estimation error of the SSL Switching algorithm presented in Algorithm 2. We choose the following minimax optimal SL and UL+ estimators

$$\hat{\boldsymbol{\theta}}_{\text{SL}} = \frac{1}{n_l}\sum_{i=1}^{n_l} Y_i X_i \tag{15}$$

$$\hat{\boldsymbol{\theta}}_{\text{UL+}} = \text{sign}\left(\hat{\boldsymbol{\theta}}_{\text{SL}}^\top \hat{\boldsymbol{\theta}}_{\text{UL}}\right)\hat{\boldsymbol{\theta}}_{\text{UL}}, \text{ with } \hat{\boldsymbol{\theta}}_{\text{UL}} = \sqrt{(\hat{\lambda}-1)_+}\hat{v}, \tag{16}$$

where $(\hat{\lambda}, \hat{v})$ is the leading eigenpair of the sample covariance matrix $\hat{\Sigma} = \frac{1}{n_u}\sum_{j=0}^{n_u} X_j X_j^T$ and we use the notation $(x)_+ := \max(0, x)$. It is known that this UL estimator is minimax optimal [39]. As unsupervised learning can only obtain classifiers up to a sign permutation, it is necessary to endow the UL+ algorithm with a means to discern between $\hat{\boldsymbol{\theta}}_{\text{UL}}$ and $-\hat{\boldsymbol{\theta}}_{\text{UL}}$, as explained in Section 2.2. In

the case of the estimator introduced in Equation (16), the sign is selected so as to obtain an estimator that aligns better with the $\hat{\boldsymbol{\theta}}_{\mathrm{SL}}$. The upper bound for the expected error incurred by the UL+ estimator is given in Proposition 2.

For the SL estimator $\hat{\boldsymbol{\theta}}_{\mathrm{SL}}$, we apply standard results for Gaussian distributions to upper bound the estimation error that holds for any regime of $n$ and $d$:

$$\mathbb{E}_{\mathcal{D}_l}\left[\|\hat{\boldsymbol{\theta}}_{\mathrm{SL}} - \boldsymbol{\theta}^*\|\right] \leq \sqrt{\frac{d}{n_l}}. \tag{17}$$

Using Equation (17) and Proposition 2 and switching between $\hat{\boldsymbol{\theta}}_{\mathrm{SL}}$ and $\hat{\boldsymbol{\theta}}_{\mathrm{UL+}}$ according to the conditions in Algorithm 2 that pick the better performing of the two algorithms depending on the regime, we can show that there exist universal constants $C, c_0 > 0$ such that for $0 \leq s \leq 1$, $d \geq 2$ and $n_u \geq (160/s)^2 d$, it holds that

$$\mathbb{E}\left[\|\hat{\boldsymbol{\theta}}_{\mathrm{SSL\text{-}S}} - \boldsymbol{\theta}^*\|\right] \leq C \min\left\{s, \sqrt{\frac{d}{n_l}}, \sqrt{\frac{d}{s^2 n_u}} + s e^{-\frac{1}{2}n_l s^2\left(1 - c_0\sqrt{\frac{d\log(n_u)}{s^2 n_u}}\right)^2}\right\}, \tag{18}$$

where the expectation is over $\mathcal{D}_l \sim \left(P_{XY}^{\boldsymbol{\theta}^*}\right)^{n_l}$ and $\mathcal{D}_u \sim \left(P_X^{\boldsymbol{\theta}^*}\right)^{n_u}$.

**Matching lower and upper bound.** When $n_l > O(\frac{\log(n_u)}{s^2})$, the exponential term becomes negligible and the first additive component dominates in the last term in the right-hand side of Equation (18). Basic calculations then yield that the expected error of the switching algorithm is upper bounded by $C' \min\left\{s, \sqrt{\frac{d}{n_l + s^2 n_u}}\right\}$ for some constant $C'$, which concludes the proof of the theorem.

## C    Proof of SSL Minimax Rate for Excess Risk in Theorem 1

In this section, we prove the minimax lower bound on excess risk for an algorithm that uses both labeled and unlabeled data and a matching (up to logarithmic factors) upper bound.

### C.1    Proof of excess risk lower bound

We first prove the excess error minimax lower bound in Theorem 1, namely we aim to show that there exists a constant $C_0 > 0$ such that for any $s > 0$, $n_u, n_l \geq 0$ and $d \geq 4$, we have

$$\inf_{\mathcal{A}_{\mathrm{SSL}}} \sup_{\|\boldsymbol{\theta}^*\|=s} \mathbb{E}\left[\mathcal{E}\left(\mathcal{A}_{\mathrm{SSL}}\left(\mathcal{D}_l, \mathcal{D}_u\right), \boldsymbol{\theta}^*\right)\right] \geq C_0 e^{-s^2/2} \min\left\{\frac{d}{s n_l + s^3 n_u}, s\right\}, \tag{19}$$

where the expectation is over $\mathcal{D}_l \sim \left(P_{XY}^{\boldsymbol{\theta}^*}\right)^{n_l}$ and $\mathcal{D}_u \sim \left(P_X^{\boldsymbol{\theta}^*}\right)^{n_u}$. Our approach to proving this lower bound is again to apply Fano's method [19] using the excess risk as the evaluation method. The reduction from estimation to testing usually hinges on the triangle inequality in a metric space. Since the excess risk does not satisfy the metric axioms, we employ a technique introduced in Azizyan et al. [2] to derive an alternative sufficient condition for applying Fano's inequality.

Let $\boldsymbol{\theta}_1, \ldots, \boldsymbol{\theta}_M \in \Theta$, $M \geq 2$, and $\gamma > 0$. If for all $1 \leq i \neq j \leq M$ and $\hat{\boldsymbol{\theta}}$,

$$\mathcal{E}\left(\hat{\boldsymbol{\theta}}, \boldsymbol{\theta}_i\right) < \gamma \quad \text{implies} \quad \mathcal{E}\left(\hat{\boldsymbol{\theta}}, \boldsymbol{\theta}_j\right) \geq \gamma, \tag{20}$$

then

$$\inf_{\mathcal{A}_{\mathrm{SSL}}} \max_{i \in [0..M]} \mathbb{E}\left[\mathcal{E}\left(\mathcal{A}_{\mathrm{SSL}}(\mathcal{D}_l, \mathcal{D}_u), \boldsymbol{\theta}_i\right)\right] \tag{21}$$

$$\geq \gamma\left(1 - \frac{1 + n_l \max\limits_{i \neq j} D\left(P_{XY}^{\boldsymbol{\theta}_i} \| P_{XY}^{\boldsymbol{\theta}_j}\right) + n_u \max\limits_{i \neq j} D\left(P_X^{\boldsymbol{\theta}_i} \| P_X^{\boldsymbol{\theta}_j}\right)}{\log(M)}\right),$$

where the expectation is over $\mathcal{D}_l \sim \left( P_{XY}^{\boldsymbol{\theta}_i} \right)^{n_l}$ and $\mathcal{D}_u \sim \left( P_X^{\boldsymbol{\theta}_i} \right)^{n_u}$.

In order to get a lower bound, we again pick $\boldsymbol{\theta}_i, \dots, \boldsymbol{\theta}_M$ to be an appropriate packing, so that the condition in Equation (20) can be satisfied. For this purpose, we can simply use the construction from Li et al. [24], which was previously used to obtain tight bounds for supervised and unsupervised settings. Let $p = (d-1)/6$. By Lemma 4.10 in Massart [26], there exists a set $\tilde{\mathcal{M}} = \{\psi_1, \dots, \psi_M\}$, such that $\|\psi_i\|_0 = p$, $\psi_i \in \{0,1\}^{d-1}$, the Hamming distance $\delta(\psi_i, \psi_j) > p/2$ for all $1 \le i < j \le M = |\tilde{\mathcal{M}}|$, and $\log M \ge \frac{p}{5} \log \frac{d}{p} \ge d \log(6)/60 = c_1 d$.

Define
$$\mathcal{M} = \left\{ \boldsymbol{\theta}_i = \begin{bmatrix} \sqrt{s^2 - p\alpha^2} \\ \alpha\psi_i \end{bmatrix} \middle| \psi_i \in \tilde{\mathcal{M}} \right\}$$

for some absolute constant $\alpha$. Note that since $\|\boldsymbol{\theta}_i\| = s$ and $\|\boldsymbol{\theta}_i - \boldsymbol{\theta}_j\|^2 = \alpha^2 \delta(\psi_i, \psi_j)$, we have

$$\frac{p\alpha^2}{2} \le \|\boldsymbol{\theta}_i - \boldsymbol{\theta}_j\|^2 \le 2p\alpha^2 \tag{22}$$

and

$$s^2 - p\alpha^2 \le \boldsymbol{\theta}_i^\top \boldsymbol{\theta}_j \le s^2 - p\alpha^2/4. \tag{23}$$

First, we show that the excess risk satisfies the condition in Equation (20). As in the proof of Theorem 1 in Li et al. [24], we have that for any $\boldsymbol{\theta}$,

$$\mathcal{E}(\boldsymbol{\theta}, \boldsymbol{\theta}_i) + \mathcal{E}(\boldsymbol{\theta}, \boldsymbol{\theta}_j) \ge 2c_0 e^{-s^2/2} \frac{p\alpha^2}{s},$$

and thus, for all $i$ and $j \ne i$, it holds that

$$\mathcal{E}(\boldsymbol{\theta}, \boldsymbol{\theta}_i) \le c_0 e^{-s^2/2} \frac{p\alpha^2}{s} \implies \mathcal{E}(\boldsymbol{\theta}, \boldsymbol{\theta}_j) \ge c_0 e^{-s^2/2} \frac{p\alpha^2}{s}. \tag{24}$$

Then since the condition in Equation (20) is satisfied, we obtain

$$\inf_{\mathcal{A}_{\text{SSL}}} \sup_{\|\boldsymbol{\theta}^*\|=s} \mathbb{E}_{\mathcal{D}_l, \mathcal{D}_u} \left[ \mathcal{E}\left( \mathcal{A}_{\text{SSL}}(\mathcal{D}_l, \mathcal{D}_u), \boldsymbol{\theta}^* \right) \right]$$
$$\ge \inf_{\mathcal{A}_{\text{SSL}}} \max_{i \in [0..M]} \mathbb{E}\left[ \mathcal{E}\left( \mathcal{A}_{\text{SSL}}(\mathcal{D}_l, \mathcal{D}_u), \boldsymbol{\theta}_i \right) \right]$$
$$\ge c_0 e^{-s^2/2} \frac{p\alpha^2}{s} \left( 1 - \frac{1 + n_l \max_{i \ne j} D\left( P_{XY}^{\boldsymbol{\theta}_i} \| P_{XY}^{\boldsymbol{\theta}_j} \right) + n_u \max_{i \ne j} D\left( P_X^{\boldsymbol{\theta}_i} \| P_X^{\boldsymbol{\theta}_j} \right)}{\log(M)} \right). \tag{25}$$

Next, we bound the KL divergences between the two joint distributions and between the two marginals that appear in Equation (25). For the joint distributions we have that:

$$D\left( P_{XY}^{\boldsymbol{\theta}_i} \| P_{XY}^{\boldsymbol{\theta}_j} \right) = \frac{1}{2} \|\boldsymbol{\theta}_i - \boldsymbol{\theta}_j\|_2^2 \le p\alpha^2, \tag{26}$$

where the inequality follows from Equation (22). Using Proposition 24 in Azizyan et al. [2], we bound the KL divergence between the two marginals

$$D\left( P_X^{\boldsymbol{\theta}_i} \| P_X^{\boldsymbol{\theta}_j} \right) \lesssim s^4 \left( 1 - \frac{\boldsymbol{\theta}_i^\top \boldsymbol{\theta}_j}{\|\boldsymbol{\theta}_i\| \|\boldsymbol{\theta}_j\|} \right) \le ps^2 \alpha^2, \tag{27}$$

where the inequality follows from (23). Plugging (26) and (27) into (25) and setting

$$\alpha^2 = c_3 \min \left\{ \frac{c_1 d - \log 2}{8(pn_l + s^2 p n_u)}, \frac{s^2}{p} \right\},$$

gives the desired result

$$\inf_{\mathcal{A}_{\text{SSL}}} \sup_{\|\boldsymbol{\theta}^*\|=s} \mathbb{E}_{\mathcal{D}_l, \mathcal{D}_u} \left[ \mathcal{E}\left( \mathcal{A}_{\text{SSL}}(\mathcal{D}_l, \mathcal{D}_u), \boldsymbol{\theta}^* \right) \right] \gtrsim e^{-s^2/2} \min \left\{ \frac{d}{sn_l + s^3 n_u}, s \right\}.$$

## C.2 Proof of excess risk upper bound

Next, we prove the upper bound on the excess risk of the SSL switching estimator $\hat{\boldsymbol{\theta}}_{\text{SSL-S}}$ output by Algorithm 2 with the supervised and unsupervised estimators defined in Appendix B.2 to show the tightness of the excess risk lower bound in Theorem 1. In particular, we show that there exist universal constants $C, c_0 > 0$ such that for $0 \leq s \leq 1$, $d \geq 2$ and for sufficiently large $n_u$ and $n_l$,

$$\mathbb{E}\left[\mathcal{E}(\hat{\boldsymbol{\theta}}_{\text{SSL-S}})\right] \leq C e^{-\frac{1}{2}s^2} \min\left\{ s, \frac{d\log(n_l)}{sn_l}, \frac{d\log(dn_u)}{s^3 n_u} + e^{-\frac{1}{2}s^2\left(n_l\left(1-c_0\sqrt{\frac{d\log(n_u)}{s^2 n_u}}\right)^2 - 1\right)} \right\},$$

where the expectation is over $\mathcal{D}_l \sim \left(P_{XY}^{\boldsymbol{\theta}^*}\right)^{n_l}$ and $\mathcal{D}_u \sim \left(P_X^{\boldsymbol{\theta}^*}\right)^{n_u}$.

The proof follows the same arguments as the proof presented in Appendix B.2 where we instead use excess risk upper bounds for SL and UL from Li et al. [24].

In addition, we also use a result that follows from Proposition 2 to choose the sign of the UL+ estimator.

Note that the upper bound on the excess risk of $\hat{\boldsymbol{\theta}}_{\text{SSL-S}}$ is matching the lower bound in Equation (19), up to logarithmic factors. We conjecture that the logarithmic factors are an artifact of the analysis and can be removed. For instance, it may be possible to extend results in Ratsaby and Venkatesh [30] that bound the excess risk using the estimation error upper bound without incurring logarithmic factors. However, their results are not directly applicable here because they are only valid under the assumption that the estimation error is arbitrarily small and hence not enough for bounding the expectation.

## D Theoretical motivation for the SSL Weighted Algorithm

In this section, we show theoretically that the SSL-W procedure introduced in Section 4 can achieve lower squared estimation error (up to sign permutation) compared to SSL-S. This result shows that it is possible to improve the error of the naïve SSL-S algorithm by making effective use of *all* the data that is available.

For the purpose of the theoretical analysis, we consider a slightly different SSL-W estimator compared to the one introduced in Section 4. First, recall that for the classification problem we consider, unsupervised learning produces a set of two feasible predictors $\{\hat{\boldsymbol{\theta}}_{\text{UL}}, -\hat{\boldsymbol{\theta}}_{\text{UL}}\}$ and cannot discern between them without access to a (small) labeled dataset. We denote by $\hat{\boldsymbol{\theta}}_{\text{UL}}^*$ the UL estimator with correct sign, namely $\hat{\boldsymbol{\theta}}_{\text{UL}}^* := \arg\min_{\boldsymbol{\theta} \in \{\hat{\boldsymbol{\theta}}_{\text{UL}}, -\hat{\boldsymbol{\theta}}_{\text{UL}}\}} \mathbb{E}\left[\|\boldsymbol{\theta} - \boldsymbol{\theta}^*\|^2\right]$. Here, the hat symbol indicates that a finite sample is used to obtain the set of estimators $\{\hat{\boldsymbol{\theta}}_{\text{UL}}, -\hat{\boldsymbol{\theta}}_{\text{UL}}\}$, while the star symbol indicates that oracle information is employed to choose the sign of the estimator.

In what follows, we study theoretically the error of the SSL-W estimator constructed using $\hat{\boldsymbol{\theta}}_{\text{UL}}^*$, i.e. $\hat{\boldsymbol{\theta}}_{\text{SSL-W}}^*(t) := t\hat{\boldsymbol{\theta}}_{\text{SL}} + (1-t)\hat{\boldsymbol{\theta}}_{\text{UL}}^*$. Therefore, our result characterizes the error of the SSL-W estimator up to a sign permutation. To choose the correct sign, one needs only a small labeled dataset, similar in size to what is prescribed by Proposition 2. While this step is not captured by Proposition 3, SSL-S is unlikely to close the gap to SSL-W when provided with this small amount of additional labeled data. Moreover, for a fairer comparison we also give the switching estimator access to the UL estimator with correct sign and define $\hat{\boldsymbol{\theta}}_{\text{SSL-S}}^* := \arg\min_{\boldsymbol{\theta} \in \{\hat{\boldsymbol{\theta}}_{\text{SL}}, \hat{\boldsymbol{\theta}}_{\text{UL}}^*\}} \mathbb{E}\left[\|\boldsymbol{\theta} - \boldsymbol{\theta}^*\|^2\right]$

We can now state Proposition 3, which shows that there exists an optimal weight for which the SSL-W predictor achieves lower estimation error than the SSL Switching predictor, $\hat{\boldsymbol{\theta}}_{\text{SSL-S}}^*$.

**Proposition 3.** *Consider a distribution $P_{XY}^{\boldsymbol{\theta}^*} \in \mathcal{P}_{2\text{-GMM}}^{(s)}$ and let $d \geq 2$, and $n_l, n_u > 0$. Let $\hat{\boldsymbol{\theta}}_{SSL\text{-}W}^*(t^*)$ be the SSL-W estimator introduced above. Then there exists a $t^* \in (0,1)$ for which*

$$\mathbb{E}\left[\left\|\hat{\boldsymbol{\theta}}_{SSL\text{-}S}^* - \boldsymbol{\theta}^*\right\|^2\right] - \mathbb{E}\left[\left\|\hat{\boldsymbol{\theta}}_{SSL\text{-}W}^*(t^*) - \boldsymbol{\theta}^*\right\|^2\right] = \min\left\{r, \frac{1}{r}\right\} \mathbb{E}\left[\left\|\hat{\boldsymbol{\theta}}_{SSL\text{-}W}^*(t^*) - \boldsymbol{\theta}^*\right\|^2\right], \quad (28)$$

*where $r = \frac{\mathbb{E}\left[\|\hat{\boldsymbol{\theta}}_{UL}^* - \boldsymbol{\theta}^*\|^2\right]}{\mathbb{E}\left[\|\hat{\boldsymbol{\theta}}_{SL} - \boldsymbol{\theta}^*\|^2\right]}$, and the expectations are over $\mathcal{D}_l \sim \left(P_{XY}^{\boldsymbol{\theta}^*}\right)^{n_l}, \mathcal{D}_u \sim \left(P_X^{\boldsymbol{\theta}^*}\right)^{n_u}$.*

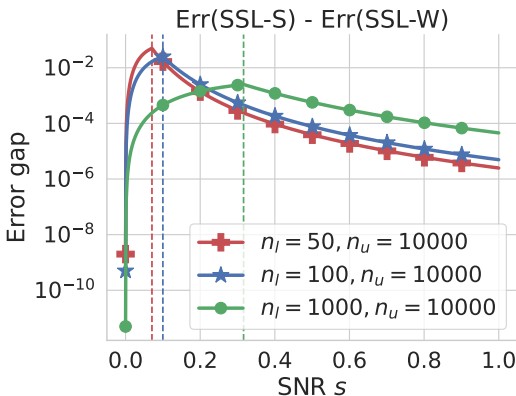

Figure 2: Estimation error gap between SSL-S and SSL-W as revealed by Proposition 3 for varying SNR and $n_l$ ($n_u = 10000$). The maximum gap is reached at the switching point, indicated by the vertical dashed lines.

Since the RHS of Equation (28) is always positive, $\hat{\boldsymbol{\theta}}^*_{\text{SSL-W}}(t^*)$ always outperforms $\hat{\boldsymbol{\theta}}^*_{\text{SSL-S}}$ as long as the conditions of Proposition 3 are satisfied. The magnitude of the error gap between SSL-S and SSL-W depends on the gap between SL and UL+ (see Figure 2). The maximum gap is reached for $\mathbb{E}\left[\left\|\hat{\boldsymbol{\theta}}^*_{\text{UL}} - \boldsymbol{\theta}^*\right\|^2\right] \approx \mathbb{E}\left[\left\|\hat{\boldsymbol{\theta}}_{\text{SL}} - \boldsymbol{\theta}^*\right\|^2\right]$ when SSL-W obtains half the error of SSL-S.

### D.1 Proof of Proposition 3

The first step in proving Proposition 3 is to express the estimation error of $\hat{\boldsymbol{\theta}}^*_{\text{SSL-W}}(t^*)$ in terms of the estimation errors of $\hat{\boldsymbol{\theta}}_{\text{SL}}$ and $\hat{\boldsymbol{\theta}}^*_{\text{UL}}$ which is captured by Lemma 1.

**Lemma 1.** *Let $\hat{\boldsymbol{\theta}}_1$ and $\hat{\boldsymbol{\theta}}_2$ be two statistically independent estimators of $\boldsymbol{\theta}^* \in \mathbb{R}^d$ and let $\hat{\boldsymbol{\theta}}_1$ be unbiased, i.e. $\mathbb{E}\left[\hat{\boldsymbol{\theta}}_1\right] = \boldsymbol{\theta}^*$. Then, the expected squared error of the weighted estimator $\hat{\boldsymbol{\theta}}_{t^*} = t^*\hat{\boldsymbol{\theta}}_1 + (1 - t^*)\hat{\boldsymbol{\theta}}_2$ with $t^* = \frac{\mathbb{E}\left[\|\hat{\boldsymbol{\theta}}_2 - \boldsymbol{\theta}^*\|^2\right]}{\mathbb{E}\left[\|\hat{\boldsymbol{\theta}}_1 - \boldsymbol{\theta}^*\|^2\right] + \mathbb{E}\left[\|\hat{\boldsymbol{\theta}}_2 - \boldsymbol{\theta}^*\|^2\right]}$ is given by*

$$\mathbb{E}\left[\|\hat{\boldsymbol{\theta}}_{t^*} - \boldsymbol{\theta}^*\|^2\right] = \left(\frac{1}{\mathbb{E}\left[\|\hat{\boldsymbol{\theta}}_1 - \boldsymbol{\theta}^*\|^2\right]} + \frac{1}{\mathbb{E}\left[\|\hat{\boldsymbol{\theta}}_2 - \boldsymbol{\theta}^*\|^2\right]}\right)^{-1}.$$

We can apply Lemma 1, since $\hat{\boldsymbol{\theta}}_{\text{SL}}$ is unbiased and $\hat{\boldsymbol{\theta}}_{\text{SL}}$ and $\hat{\boldsymbol{\theta}}^*_{\text{UL}}$ are trained on $\mathcal{D}_l$ and $\mathcal{D}_u$ respectively, and hence, are independent. The proof then follows from calculating the difference between the harmonic mean and the minimum of estimation errors of $\hat{\boldsymbol{\theta}}_{\text{SL}}$ and $\hat{\boldsymbol{\theta}}_{\text{UL+}}$. Let $x, y \in \mathbb{R}_+$ and w.l.o.g. assume $x \leq y$. Then we have:

$$x - \left(\frac{1}{x} + \frac{1}{y}\right)^{-1} = x - \frac{xy}{x+y} = \frac{x^2}{x+y} = \frac{x}{y}\frac{xy}{x+y}.$$

Applying this identity concludes the proof after choosing $t^* = \frac{\mathbb{E}\left[\|\hat{\boldsymbol{\theta}}_2 - \boldsymbol{\theta}^*\|^2\right]}{\mathbb{E}\left[\|\hat{\boldsymbol{\theta}}_1 - \boldsymbol{\theta}^*\|^2\right] + \mathbb{E}\left[\|\hat{\boldsymbol{\theta}}_2 - \boldsymbol{\theta}^*\|^2\right]}$ and for $x = \min\left\{\mathbb{E}[\|\hat{\boldsymbol{\theta}}^*_{\text{UL}} - \boldsymbol{\theta}^*\|^2], \mathbb{E}[\|\hat{\boldsymbol{\theta}}_{\text{SL}} - \boldsymbol{\theta}^*\|^2]\right\}$ and $y = \max\left\{\mathbb{E}[\|\hat{\boldsymbol{\theta}}^*_{\text{UL}} - \boldsymbol{\theta}^*\|^2], \mathbb{E}[\|\hat{\boldsymbol{\theta}}_{\text{SL}} - \boldsymbol{\theta}^*\|^2]\right\}$.

**Remark.** Note that this lemma holds for arbitrary distributions and estimators as long as they are independent and one of them is unbiased. Therefore, future results that derive upper bounds for SL and UL+ for other distributional assumptions and estimators can seamlessly be plugged into Lemma 1. By the same argument, $\hat{\boldsymbol{\theta}}_{\text{SSL-W}}$ obtained by other SL and UL+ estimators can also be expected to improve over the respective SL and UL+ estimators, given that one of them is unbiased.

## D.2 Proof of Lemma 1

By definition of $\hat{\boldsymbol{\theta}}_{t^*}$, we have

$$
\begin{aligned}
\mathbb{E}\left[\|\hat{\boldsymbol{\theta}}_{t^*} - \boldsymbol{\theta}^*\|^2\right] &= \mathbb{E}\left[\|t^*\hat{\boldsymbol{\theta}}_1 + (1 - t^*)\hat{\boldsymbol{\theta}}_2 - \boldsymbol{\theta}^*\|^2\right] \\
&= \mathbb{E}\left[t^{*2}\|\hat{\boldsymbol{\theta}}_1 - \boldsymbol{\theta}^*\|^2 + (1 - t^*)^2\|\hat{\boldsymbol{\theta}}_2 - \boldsymbol{\theta}^*\|^2 + 2t^*(1 - t^*)(\hat{\boldsymbol{\theta}}_1 - \boldsymbol{\theta}^*)^\top(\hat{\boldsymbol{\theta}}_2 - \boldsymbol{\theta}^*)\right] \\
&= \mathbb{E}\left[t^{*2}\|\hat{\boldsymbol{\theta}}_1 - \boldsymbol{\theta}^*\|^2 + (1 - t^*)^2\|\hat{\boldsymbol{\theta}}_2 - \boldsymbol{\theta}^*\|^2\right],
\end{aligned}
$$

where the last equality holds due to the independence of $\hat{\boldsymbol{\theta}}_1$ and $\hat{\boldsymbol{\theta}}_2$ and the unbiasedness of $\hat{\boldsymbol{\theta}}_1$.

Plugging in $t^* = \frac{\mathbb{E}\|\hat{\boldsymbol{\theta}}_2 - \boldsymbol{\theta}^*\|^2}{\mathbb{E}\|\hat{\boldsymbol{\theta}}_1 - \boldsymbol{\theta}^*\|^2 + \mathbb{E}\|\hat{\boldsymbol{\theta}}_2 - \boldsymbol{\theta}^*\|^2}$, we get

$$
\begin{aligned}
\mathbb{E}\|\hat{\boldsymbol{\theta}}_{t^*} - \boldsymbol{\theta}^*\|^2 &= \left(\frac{\mathbb{E}\|\hat{\boldsymbol{\theta}}_2 - \boldsymbol{\theta}^*\|^2}{\mathbb{E}\|\hat{\boldsymbol{\theta}}_1 - \boldsymbol{\theta}^*\|^2 + \mathbb{E}\|\hat{\boldsymbol{\theta}}_2 - \boldsymbol{\theta}^*\|^2}\right)^2 \mathbb{E}\|\hat{\boldsymbol{\theta}}_1 - \boldsymbol{\theta}^*\|^2 \\
&\quad + \left(\frac{\mathbb{E}\|\hat{\boldsymbol{\theta}}_1 - \boldsymbol{\theta}^*\|^2}{\mathbb{E}\|\hat{\boldsymbol{\theta}}_1 - \boldsymbol{\theta}^*\|^2 + \mathbb{E}\|\hat{\boldsymbol{\theta}}_2 - \boldsymbol{\theta}^*\|^2}\right)^2 \mathbb{E}\|\hat{\boldsymbol{\theta}}_2 - \boldsymbol{\theta}^*\|^2 \\
&= \frac{\mathbb{E}\|\hat{\boldsymbol{\theta}}_1 - \boldsymbol{\theta}^*\|^2 \mathbb{E}\|\hat{\boldsymbol{\theta}}_2 - \boldsymbol{\theta}^*\|^2}{\mathbb{E}\|\hat{\boldsymbol{\theta}}_1 - \boldsymbol{\theta}^*\|^2 + \mathbb{E}\|\hat{\boldsymbol{\theta}}_2 - \boldsymbol{\theta}^*\|^2} \\
&= \frac{1}{\frac{1}{\mathbb{E}\|\hat{\boldsymbol{\theta}}_1 - \boldsymbol{\theta}^*\|^2} + \frac{1}{\mathbb{E}\|\hat{\boldsymbol{\theta}}_2 - \boldsymbol{\theta}^*\|^2}}.
\end{aligned}
$$

# E  Simulation details

## E.1  Methodology

We split each dataset in a test set, a validation set and a training set. The unlabeled set size is fixed to 5000 for the synthetic experiments and 4000 for the real-world datasets. The size of the labeled set $n_l$ is varied in each experiment. For each dataset, we draw a different labeled subset 20 times and report the average and the standard deviation of the error gap (or the error) over these runs. The (unlabeled) validation set and the test set have 1000 labeled samples each.

We use logistic regression from Scikit-Learn [28] as the supervised algorithm. We use the validation set to select the ridge penalty for SL. For the unsupervised algorithm, we use an implementation of Expectation-Maximization from the Scikit-Learn library. We also use the self-training algorithm from Scikit-Learn with a logistic regression estimator. The best confidence threshold for the pseudolabels is selected using the validation set. Moreover, the optimal weight for SSL-W is also chosen with the help of the validation set. Since we use an unlabeled validation set, we need to employ an appropriate criterion for hyperparameter selection. Therefore, we select models that lead to a large (average) margin measured on the unlabeled validation set. We give SSL-S the benefit of choosing the optimal switching point between SL and UL+ by using the test set. Even with this important advantage, SSL-W (and sometimes self-training) still manage to outperform SSL-S.

## E.2  Details about the real-world datasets

**Tabular data.**   We select tabular datasets from the OpenML repository [37] according to a number of criteria. We focus on high-dimensional data with $100 \le d < 1000$, where the two classes are not suffering from extreme class imbalance, i.e. the imbalance ratio between the majority and the minority class is at most 5. Moreover, we only consider datasets that have substantially more samples than the number of features, i.e. $\frac{n}{d} > 10$. In the end, we are left with 5 datasets, that span a broad range of application domains, from ecology to chemistry and finance.

To assess the difficulty of the datasets, we train logistic regression on the entire data that is available, and report the training error. Since we train on substantially more samples than the number of

dimensions, the predictor that we obtain is a good estimate of the linear Bayes classifier for each dataset.

Furthermore, we measure the extent to which the data follows a GMM distribution with spherical components. We fit a spherical Gaussian to data coming from each class and use linear discriminant analysis (LDA) for prediction. We record the training error (of the best permutation). Intuitively, this is a score of how much our assumption about the connection between the marginal $P_X$ and the labeling function $P(Y|X)$ is satisfied. For Figure 1 we take $\boldsymbol{\theta}^*_{\text{Bayes}}$ to be the linear Bayes classifier and $\boldsymbol{\theta}^*_{UL}$ the LDA classifier described above.[5] If the data is almost linearly separable (i.e. $\mathcal{R}_{\text{pred}}(\boldsymbol{\theta}^*_{\text{Bayes}}) \leq 0.01$), then we simply take the linear Bayes error as the compatibility.

**Image data.** In addition to the tabular data, we also consider a number of datasets that are subsets of the MNIST dataset [23]. More specifically, we create binary classification problems by selecting class pairs from MNIST. We choose 5 classification problems which vary in difficulty, as measured by the Bayes error, from easier (e.g. digit 0 vs digit 1) to more difficult (e.g. digit 5 vs digit 9). Table 3 presents the exact class pairs that we selected. To make the problem more amenable for linear classification, we consider as covariates the 20 principle components of the MNIST images.

| Dataset name | $d$ | Linear classif. training error | LDA w/ spherical GMM training error |
|---|---|---|---|
| mnist_0v1 | 784 | 0.001 | 0.009 |
| mnist_1v7 | 784 | 0.006 | 0.036 |
| madeline | 259 | 0.344 | 0.381 |
| philippine | 308 | 0.240 | 0.318 |
| vehicleNorm | 100 | 0.141 | 0.177 |
| mnist_5v9 | 784 | 0.024 | 0.045 |
| mnist_5v6 | 784 | 0.024 | 0.042 |
| a9a | 123 | 0.150 | 0.216 |
| mnist_3v8 | 784 | 0.042 | 0.105 |
| musk | 166 | 0.037 | 0.270 |

Table 3: Some characteristics of the datasets considered in our experimental study.

# F   More experiments

In this section we present further experiments that indicate that the SSL Weighted algorithm (SSL-W) can indeed outperform the minimax optimal Switching algorithm (SSL-S). Figures 3 and 4 show that there exists a positive error gap between SSL-S and SSL-W for a broad range of $n_l$ values, for synthetic and real-world data, respectively. The extent of the error gap is determined by the $\frac{n_u}{n_l}$ ratio as well as the signal-to-noise ratio that is specific to each data distribution. In addition, Figure 4 also indicates that self-training can outperform SSL-W in some scenarios. It remains an exciting direction for future work to provide a tight analysis of self-training that can indicate when it outperforms both SL and UL+ simultaneously.

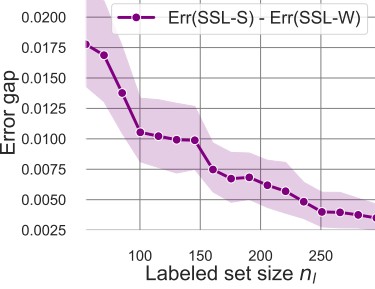

Figure 3: Error gap between SSL-S and SSL-W on synthetic 2-GMM data with $s = 0.1$. The positive gap indicates that SSL-W outperforms SSL-S (and hence, also SL and UL+).

---

[5]Note that we refer to the LDA estimator as *UL* since we use it as a proxy to assess how well unsupervised learning can perform on each dataset.

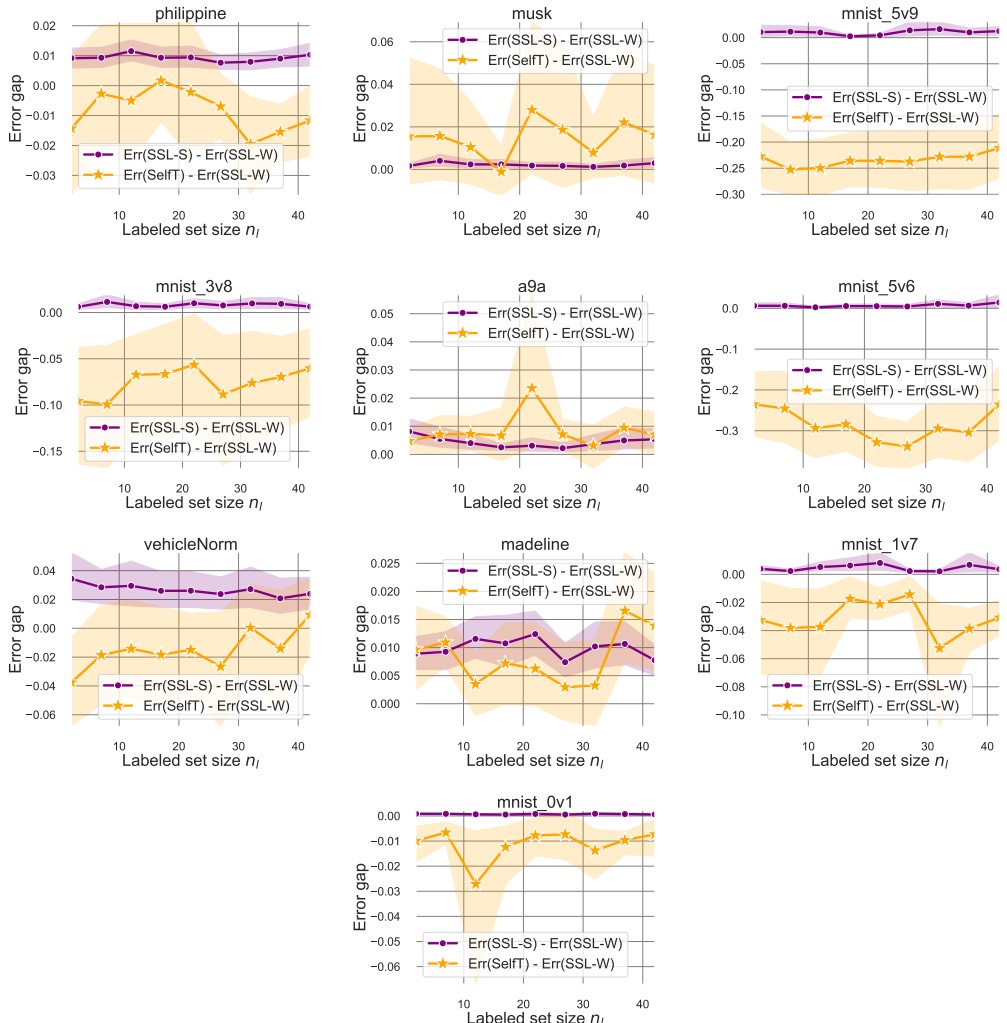

Figure 4: Error gap between SSL-S/self-training and SSL-W on real-world datasets. The positive gap indicates that SSL-W (and, in turn, self-training) outperforms SSL-S (and hence, also SL and UL+) for a broad range of $n_l$ values.

