# OpenReview forum: "Can semi-supervised learning use all the data effectively? A lower bound perspective"
_NeurIPS.cc/2023/Conference — NeurIPS 2023 spotlight_

### Official Review · Reviewer_9F3v · 2023-07-05

**Soundness:** 3 good
**Presentation:** 3 good
**Contribution:** 4 excellent
**Rating:** 7
**Confidence:** 4

**Summary:**

this paper investigates whether improvements in the rates of sample complexities are possible for semi-supervised learning compared with supervised learning and unsupervised learning (up to a change in sign) for the specific case of a classification problem where each class is a symmetrical Gaussian. it is found that no improvement in the rate can be made, but it is possible to improve the constants.

**Strengths:**

1 I think it is a very relevant question to investigate whether SSL can improve performance in theory over sup and unsup, and this can have a great impact

2 I like the deep theoretical analysis where minimax upper and lower bounds are derived

3 the theory is complemented by some experiments to validate the theoretical findings

**Weaknesses:**

4 the work misses a very relevant recent survey concerning this question in general:
Mey, A., & Loog, M. (2019). Improvability through semi-supervised learning: A survey of theoretical results. arXiv preprint arXiv:1908.09574.



**Questions:**

5 for Algorithm 3, how does it make sense that $t$ is an input? should this not be learned also? similarly, I see the same problem with Theorem 6. This would imply the algorithm has knowledge of the distribution? Or is there a universal (distribution independent) t?

6 Similarly, in the experiments, these parameters seem to be tuned on a holdout labeled set. Why is that appropiate?

---

> ### Author Rebuttal · Authors · 2023-08-10
>
> We thank you for appreciating the importance of the problem as well as the thorough theoretical and experimental analyses in our paper. We are grateful for the feedback that you have provided and hope that our answers address the points raised in your review.
>
> > 5 for Algorithm 3, how does it make sense that $t$ is an input? should this not be learned also? similarly, I see the same problem with Theorem 6. This would imply the algorithm has knowledge of the distribution? Or is there a universal (distribution independent) t?
>
> The reviewer is correct in noting that the algorithm takes $t$ as a hyperparameter And that different $t$ leads to different estimators. The $t^\star$ that achieves a smaller risk than the switching algorithm via Theorem 6 indeed depends on $s$ (for both algorithms that we are comparing), which is a distributional parameter that describes the hardness of the problem. Theorem 6 belongs to a standard type of result that shows the existence of a hyper-parameter for which the estimator achieves certain favorable properties [Wei et al; Li et al, Meinshausen et al; van de Geer]. In practice, it is very common to use standard data-dependent model selection tools that also can achieve good performance. We show this empirically  and leave an analysis of the impact of model selection with held-out data as an exciting direction for future work. We refer the reviewer to the “General comments” for a more detailed discussion on this topic. For Theorem 6 specifically, we would like to furthermore emphasize that both SSL-S and SSL-W with optimal $t^\star$ require the knowledge of $s$.
>
>
>
>
> > the work misses a very relevant recent survey concerning this question in general
>
> Thank you for bringing this to our attention! As a survey paper that is aimed at the same question as us, it does indeed discuss a lot of prior work that our paper is based on. We will include this reference in our related work section in the revised version.
>
> References:
>
> [Wei et al] – Y. Wei, F. Yang, M. Wainwright. Early stopping for kernel boosting algorithms: A general analysis
> with localized complexities, 2017.
>
> [Li et al] – M. Li, M. Soltanolkotabi, S. Oymak. Gradient Descent with Early Stopping is Provably Robust to Label Noise for Overparameterized Neural Networks, 2020.
>
> [Meinshausen et al] – N. Meinshausen, P. Buhlmann. High-dimensional graphs and variable selection with the Lasso, 2006.
>
> [van de Geer] – S. van de Geer. On tight bounds for the Lasso, 2018.

---

### Official Review · Reviewer_4XfH · 2023-07-06

**Soundness:** 3 good
**Presentation:** 3 good
**Contribution:** 3 good
**Rating:** 6
**Confidence:** 3

**Summary:**

The paper presents a detailed analysis of semi-supervised learning (SSL) algorithms. Specifically, the authors establish lower bounds for 2-Gaussian mixture model distributions, revealing that no SSL algorithm can improve the sample complexities of optimal supervised or unsupervised learning. However, SSL can improve their error rates by a constant factor. The authors propose an algorithm that achieves lower error than both supervised and unsupervised learning, and conduct experiments on synthetic and real-world data.

**Strengths:**

- The paper investigates the learning capacity of SSL algorithms and theoretically establishes a lower bound, which is an inherently interesting and meaningful endeavor.

- The paper is well-organized and easy to comprehend. The authors provide a clear overview of the problem and their proposed solution.

**Weaknesses:**

- Some details about the method and theory part should be further elaborated on, see Questions below.

**Questions:**

- The paper only considers Gaussian mixture distributions. Could more complex distributions be considered in future work to accommodate more real-world problems?

- The paper only considers the worst-case scenario. Is it possible to consider milder cases? For instance, could a data-dependent bound that relates to problem difficulty be derived?

- The statement on line 103 seems to be erroneous: when using the UL algorithm, why can't labeled data be considered as unlabeled data, so that UL can use $n_l + n_u$?


**Limitations:**

- As mentioned in the Questions section, the authors should provide further clarification on the method and theory parts.

---

> ### Author Rebuttal · Authors · 2023-08-09
>
> We would like to thank you for appreciating the clarity of our manuscript and the importance of the contribution presented in it. In the “General comments” we address your question regarding possible extensions of our results to more general distributions, beyond GMMs. In what follows, we answer the remaining points in the review.
>
> > The paper only considers the worst-case scenario. Is it possible to consider milder cases? For instance, could a data-dependent bound that relates to problem difficulty be derived?
>
> We would like to point out that one of the novel characteristics of the lower bound that we derive is that it adapts to the problem difficulty, which in the setting considered in the paper is quantified by the signal-to-noise ratio $s$. This is in contrast to existing lower bounds for SSL [6, 18, 32] that only focuses on the worst-case scenario in which SSL achieves the same sample complexity as SL. We hope that our answer addresses the point that you have raised. If you have further questions regarding this aspect, we would like to kindly ask you to let us know, and we would be happy to provide further clarifications.
>
>
> > The statement on line 103 seems to be erroneous: when using the UL algorithm, why can't labeled data be considered as unlabeled data, so that UL can use n_l+n_u?
>
> Thank you for raising this point! This paragraph should indeed be revised.
>
> First of all, it is indeed true that UL+ can use the labeled data without the labels in the first stage, when it performs unsupervised learning. However, typically, in empirical papers, UL+ style algorithms (e.g. SimCLR, SwAV etc) do not use the labeled data in this way since $n_u =\omega(n_l)$, and hence, it would not make a significant difference. This fact is also reflected in our rate comparisons: even if we allow UL+ to use the samples in $\mathcal{D}_l$ as unlabeled data, only the last line of Table 1 would change by a small constant in the regime when $n_u / u(n_l) = \Theta(n_l)$. More importantly, the main takeaway from Table 1 would not change: SSL cannot be both better than UL+ and SL at the same time since there is no regime for which $h_l$ and $h_u$ are simultaneously $0$. For the above reasons and in order to reduce notational overhead, we chose the slightly simpler definition of UL+ as algorithms that do not use the labeled set as unlabeled data. However, we acknowledge that you raise a very natural question, and, if it helps to avoid confusion, we would be happy to revise the definition of UL+ to also use the labeled data without the labels.
>
> Secondly, the paragraph you have referenced (lines 103-110) is intended to show how UL+ algorithms are not making the best use of the available data. This shortcoming is not immediately obvious in some of the practical settings where UL+ style algorithms were applied in prior work, i.e. where unlabeled data is many orders of magnitude more numerous than labeled data. However, the “wasteful” aspect of UL+ algorithms becomes clear when $n_u$ is, for instance, only larger than $n_l$ by a constant factor. The issue is caused by the two-stage nature of UL+, which selects and commits to a small set of estimators using the unlabeled set, before choosing among them using labeled data.
>
> Finally, in this paragraph (lines 103-110), we consider an exaggerated scenario in order to illustrate this failure of UL+. This is, of course, not a scenario likely to occur in practical applications, where $n_u$ is usually not smaller than $n_l$. We do acknowledge, however, that the current phrasing can lead to confusion, and will change it in the final version of the manuscript to better reflect our intention.
>
> We again appreciate your time and effort in critically reviewing our paper. We will be happy to promptly answer any further questions you may have about our manuscript.

---

> > ### Comment · Reviewer_4XfH · 2023-08-16
> >
> > Thanks for your response. I don't have any further questions.

---

### Official Review · Reviewer_KqNv · 2023-07-06

**Soundness:** 3 good
**Presentation:** 3 good
**Contribution:** 3 good
**Rating:** 5
**Confidence:** 2

**Summary:**

This paper analyzes the effect of SSL methods to improve the error bound. The results suggest that SSL cannot improve over the statistical rates of both SL and UL at the same time, but it is possible to improve the errors by a constant factor. Simple experiments on synthetic and small-scale real-world data validate the theoretical findings.

**Strengths:**

- This paper studies an interesting and important problem: whether semi-supervised learning can effectively use all the data.
- A detailed theoretical study is provided on the 2-GMM distribution, which sheds light on this problem. Modification:
- Though simple, the proposed SSL switching algorithm is interesting and insightful.

**Weaknesses:**

- The theoretical analysis is based on GMM, which introduces a strong assumption about the data distribution. This limits the scope of application of the theoretical analysis.
- The proposed SSL switching algorithm introduces an extra hyperparameter s/t, and it seems that the algorithm is sensitive to the choice of the hyperparameter. Though it can be chosen by holdout labeled validation set, labeled data are relatively rare in the SSL setting. Is it possible to estimate the value of the hyperparameters?

**Questions:**

Please address the concerns in the weaknesses part.

**Limitations:**

The main limitation of this paper is that the analyses mainly focus on GMMs, which limits the scope of application.

---

> ### Author Rebuttal · Authors · 2023-08-09
>
> We would like to thank you for appreciating the importance of the problem that we study as well as the significance of the insights that follow from our theoretical analysis. We are also grateful for bringing up the important observation that in the SSL setting, model selection based on only the unlabeled data is much more desirable than using a labeled validation set. As we argue in the “General comments” above, we find that selecting hyperparameters using a margin-based criterion computed on unlabeled validation data leads to similar results as the ones currently presented in the manuscript. We provide as evidence for this claim some key experimental results that we include in the attached PDF file. We will update all experiments to use the unsupervised model selection metric and stress that this does not affect in any way the takeaways of our experimental analysis.
>
> We are thankful for your time and effort in reviewing our paper. In case you have any further questions regarding this manuscript, we will be happy to answer them promptly.

---

### Official Review · Reviewer_N4Bc · 2023-07-07

**Soundness:** 4 excellent
**Presentation:** 4 excellent
**Contribution:** 4 excellent
**Rating:** 8
**Confidence:** 3

**Summary:**

The paper provides a tight lower bound for semi-supervised learning for the 2GMM model. It compares the minimax rate with supervised and unsupervised learning. The authors also provide supporting experiments on real-world and synthetic datasets."

**Strengths:**

1. The paper is very well-written. The authors have effectively communicated their ideas, making it easy for readers to follow and understand the research.

2. The results presented in the paper go beyond existing prior work (when s > 0).

3. The authors have conducted experiments to support their theoretical analysis. This empirical validation strengthens the credibility of their findings and enhances the practical relevance of the proposed approach.

**Weaknesses:**

The paper relies on strong assumptions regarding the 2GMM and linear model. This might limit the generalizability of the proposed method to other scenarios or data distributions.

**Questions:**

1. Line 65: There seems to be a typographical error in the notation "A(D_l, D_l)." Please clarify this notation.

2. Line 121: I assume that 'u' represents a function mapping the number of labeled data points to the corresponding number of unlabeled data points. Can you please confirm this understanding?

**Limitations:**

The authors have adequately addressed the limitations.

---

> ### Author Rebuttal · Authors · 2023-08-09
>
> Thank you for appreciating our results and empirical validation as well as the clarity of our writing! We are grateful for the points that you have brought up in the review! We addressed your comment on the 2-GMM distribution in the “General comments” and now answer your remaining specific questions.
>
> > There seems to be a typographical error in the notation "A(D_l, D_l)." Please clarify this notation.
>
> Indeed, the notation should read $\mathcal{A}(\mathcal{D}_I, \mathcal{D}_u)$, that is, the algorithm $\mathcal{A}$ is applied to the labeled set $\mathcal{D}_l$ and the unlabeled set $\mathcal{D}_u$ – we will correct this and other typos in the final version of our manuscript.
>
>
> > I assume that 'u' represents a function mapping the number of labeled data points to the corresponding number of unlabeled data points. Can you please confirm this understanding?
>
> Your understanding is indeed correct.
>
> We again appreciate your time and effort in reviewing the paper.  We will be happy to answer any further questions you may have about our manuscript.

---

> > ### Comment · Reviewer_N4Bc · 2023-08-15
> >
> > I have read the authors' rebuttal. I feel they have adequately addressed my questions and concerns. I am leaning toward leaving my score as it is.

---

### Author Rebuttal · Authors · 2023-08-09

We extend our sincere thanks to the reviewers for their thorough review of our manuscript and for the constructive feedback provided. We are heartened by the acknowledgment of the importance of our research question: whether SSL algorithms can enhance performance over optimal SL and UL algorithms. It is particularly encouraging that all reviewers considered our theoretical analyses to be "inherently interesting" and that it "goes beyond prior work". Furthermore, we are gratified that our experimental analysis, which "validates our theory", was deemed “insightful” by the reviewers.

We deeply appreciate the feedback and recognize its helpful role in improving our manuscript. In the ensuing comments, we address the prominent concerns raised in the reviews. Specific feedback from the reviewers will be attended to in our individual responses.

**Theoretical analysis focuses solely on GMM distributions.**

We agree that it would be interesting to obtain theoretical results that go beyond GMM distributions in future work. Nevertheless, we would like to emphasize the following:

 Analyzing the lower bound for GMM distributions is insightful in its own right:

Not only is it common to model various real-world distributions as a mixture of Gaussians [Bouguila et al], but this choice also allows our analysis to yield precise upper and lower bounds that depend on the hardness of the problem. In particular, this is reflected by the dependence on the signal-to-noise ratio $s$ in our results.

Previous works on SSL with concrete upper bounds have also focused on 2-GMMs (or slight generalizations thereof) for likely similar reasons (e.g. [16]). Moreover, giving lower or upper bounds for this commonly analyzed setting has the added benefit that it allows for comparisons with prior works.

Going beyond GMM distributions requires fundamental advances in the analysis of UL algorithms, which we consider to be beyond the scope of this paper. For instance, it is only recently that the (tight) lower and upper bounds for UL on isotropic 2-GMMs have been proven [12, 13, 22, 35], and precise rates for more general distributions are even more difficult to come by.

Having said that, we agree with reviewers that analysis of more general distributions would be interesting and leave that to future work.

**Algorithms 1, and 2 take in the switching point/weighting coefficient as parameters.**

The primary goal of this paper is to prove a “hardness”-dependent minimax lower bound and show that it is tight. For the purpose of the latter, we prove a simple upper bound for the SSL-S algorithm (with oracle knowledge of $s$). As a second contribution, we prove that there exist algorithms that can improve over the error of SSL-S by a constant factor and give one example (SSL-W with a particular choice of $t$) – papers of such flavor are common, e.g. the theoretical analyses of early stopping [Wei et al; Li et al], or Lasso [Meinshausen et al; van de Geer].

Even though our optimality guarantees for the algorithms hold for specific hyperparameters $s$ and $t$ that are derived from theory and unknown to the method, in practice, we can estimate them using held-out data, which is what we show empirically in our experiments (see also the next section of the “General comments” about labeled vs. unlabeled validation data).

We agree that it would be desirable to prove guarantees for data-dependent hyperparameter search with an appropriate model selection procedure. We believe this is one of the many exciting directions for future work that could follow from our paper.

**In experiments, the switching point and weighting coefficient are selected using labeled data, which is scarce in SSL settings.**

We thank the reviewers for bringing up this point which is indeed important for applying the algorithms in practice. We have now run additional experiments using unsupervised validation data for model selection and observe the same trends as when using a labeled validation set. We use a margin-based metric for hyperparameter selection i.e. we choose the hyperparameters that lead to the model with the largest (average) margin on the unlabeled validation set. We attach the revised version of Figures 3a and 3b in the PDF file and will change all experiments to use an unlabeled validation set. Notably, the new figures preserve the important trends that we discuss in Section 4.2, namely: 1) SSL-W is always better than SL and UL+; and 2) the gap between SSL-W and SL decreases with the SNR, and the gap between SSL and UL+ increases with the SNR.

References:

[Bouguila et al] – N. Bouguila and W. Fan. Mixture Models and Applications, 2019.

[Wei et al] – Y. Wei, F. Yang, M. Wainwright. Early stopping for kernel boosting algorithms: A general analysis
with localized complexities, 2017.

[Li et al] – M. Li, M. Soltanolkotabi, S. Oymak. Gradient Descent with Early Stopping is Provably Robust to Label Noise for Overparameterized Neural Networks, 2020.

[Meinshausen et al] – N. Meinshausen, P. Buhlmann. High-dimensional graphs and variable selection with the Lasso, 2006.

[van de Geer] – S. van de Geer. On tight bounds for the Lasso, 2018.

---

### Decision · Program_Chairs · 2023-09-21

**Decision:**

Accept (spotlight)

**Comment:**

Unanimous accept from reviewers, congrats to the authors for an excellent paper.